# Decoupling Dependency Structures: Sklar's Theorem for Explainable Outlier Detection

## Abstract

Recent advances in outlier detection have been primarily driven by deep learning models, which, while powerful, have substantial drawbacks in terms of explainability. This is particularly relevant in fields that demand detailed reasoning and understanding of why observations are classified as outliers. To close the gap between state-of-the-art performance and enhanced explainability, we propose Vine Copula-Based Outlier Detection (VC-BOD). We utilize Sklar's theorem in conjunction with vine copulas and univariate kernel density estimators to decouple marginal distributions and their dependency structure for outlier detection. Our model uses a closed-form equation for the outlier score, which allows for detailed explainability and feature attribution. VC-BOD employs a traceable criterion to determine whether a new observation is an outlier, while also identifying the specific features responsible for this classification. The proposed model further distinguishes whether these features deviate from their own distributions or from interactions with other features. Our empirical evaluations demonstrate that VC-BOD outperforms most benchmarked classical models and several deep learning approaches in terms of average rank performance while proving competitive with the best-performing models.

## 1 Introduction

Outlier detection (OD), or anomaly detection (AD), has experienced substantial advances, primarily fueled by the emergence of modern deep learning architectures and the increased availability of computational resources. These advancements have been particularly notable in the fields of computer vision and natural language processing (Kim et al., 2020; Liznerski et al., 2021). More recently, deep models for tabular data have also gained popularity (Thimonier et al., 2024; Shenkar & Wolf, 2022). While these developments have significantly pushed the boundaries of OD research in tabular data, they concurrently introduce challenges, concerning interpretability, explainability, and reliability—attributes critical in many practical applications (Hilal et al., 2022; Malaiya et al., 2018). In contrast, traditional OD methods, such as KNN (Ramaswamy et al., 2000), Isolation-Forest (IForest) (Hariri et al., 2021) and COPOD (Li et al., 2020), which do not rely on Multi-Layer Perceptrons (MLPs), can offer notable advantages regarding interpretability, computational efficiency, and explainability. However, this field has not been the main focus of recent research, with potential benefits left unexplored.

The most straightforward approach to detect outliers is to estimate the complete distribution of the data and then determine the density of a new observation. If, for a new observation, the density is strikingly small, it is deemed anomalous. However, this becomes problematic with high-dimensional data (Zimek et al., 2012). To circumvent this problem, Li et al. (2020; 2023) have demonstrated that univariate empirical cumulative distribution functions can effectively model marginal distributions for outlier detection. Yet, these models make several restrictive distributional assumptions and put little emphasis on the interactions between features, which can be crucial for understanding more complex anomaly patterns (Ahirwar et al., 2012). Horváth et al. (2020) aim to estimate the complete distribution by explicitly modeling the dependency structure between certain features, utilizing a composition of bivariate copulas. Yet, neither of these approaches utilize Sklar's Theorem (Sklar, 1959) to its full potential, which states that any multivariate distribution can be decomposed into its marginal distributions and a copula function that captures the dependency structure. Horváth et al. (2020) approximate the dependency structure to a limited degree, they only considers two features

at a time and do not separate the marginals from their dependency structure. The former limitation hinders the detection of more complex dependencies, while the latter reduces explainability.

Building on this foundation, we introduce **V**ine **C**opula-**B**ased **O**utlier **D**etection (VC-BOD), to bridge the gap between recent state-of-the-art models' high performance and a theoretically motivated understanding of outlier scores. To improve on the shortcomings detailed in the previous paragraph, we fully utilize Sklar's theorem in conjunction with vine copulas (Joe, 1997; Bedford & Cooke, 2002; Aas et al., 2009) and univariate kernel methods (Marron & Wand, 1992; Sheather & Jones, 1991). Vine copulas are multivariate distributions, constructed out of several bivariate copulas which are arranged across successive tree structures. A vine copula is a highly flexible model capable of capturing diverse dependence patterns. We construct an outlier detector by aggregating separately calculated scores for the marginal distributions and each feature interaction, as captured by the vine copula, in a closed-form equation. The copulas appearing in the higher trees of the vine copula enable us to consider multiple features together, thus capturing more complex dependencies. We leverage the individual scores to determine those features which significantly contribute to an observation being classified as an outlier. Furthermore, for each influential feature, we can determine whether it was flagged due to a deviation from its marginal distribution or because of its interactions with other features. This degree of feature attribution establishes a new benchmark in the field.

To clarify the process behind our approach, we provide a theoretical example based on a synthetic dataset, that showcases the workings of VC-BOD. To evaluate the performance we follow the well-established experimental design from works such as Thimonier et al. (2024); Shenkar & Wolf (2022), and conduct a comprehensive analysis across a set of 31 tabular datasets to benchmark VC-BOD against existing OD methodologies. The empirical findings confirm that our model consistently demonstrates state-of-the-art performance, achieving the highest average rank among non-MLP-based models and the third-highest rank across all models. A statistical test further indicates that there is no significant gap between VC-BOD and the best-performing baseline models. VC-BOD is thus to be positioned at the forefront of OD methodologies in terms of performance, while simultaneously offering one of the most advanced explainability frameworks for outlier detection.

To summarize, our work offers the following contributions:

1. A new OD-method, theoretically motivated and defined via a closed-form equation.
2. A new framework for enhanced explainability and detailed feature attribution.
3. State-of-the-art outlier detection capabilities.

## 2 RELATED WORK

For the contexts of our work, we categorize OD methodologies into two main categories: classical, which encompass models that do not utilize MLPs, and MLP-based. This distinction is important because MLP-based approaches often suffer from a unique set of limitations that are not present with classical approaches, as we discuss in the subsequent paragraphs. VC-BOD is to be classified under the distribution-based models of the classical category.

**Classical.** OD approaches without MLPs can be broadly classified into four main categories: distribution-based, proximity-based, reconstruction-based, and one-class classification methods.

*Distribution-based* methods aim to learn the underlying distribution of the data and infer the likelihood of a sample belonging to this distribution. The early works in this category, such as Parzen (1962) and Roberts & Tarassenko (1994), laid the foundation by modeling data distributions directly. However, challenges arise when dealing with high-dimensional or multimodal data, as real-world data rarely conforms neatly to standard probability distributions. More recent methods, such as Li et al. (2020; 2023) and Horváth et al. (2020), utilize copula analysis to capture the data distribution.

*Proximity-based* methods utilize distance information in the representation space to identify anomalies. A straightforward yet effective approach was introduced by Ramaswamy et al. (2000), where outliers are classified based on their distance to their nearest neighbors. More refined methods, such as Local Outlier Factor (Breunig et al., 2000), incorporate local density information, identifying a data point as an outlier if it has a substantially lower local density compared to its neighbors. While

they have been applied with great success, Zimek et al. (2012) have shown that these methods also suffer from the curse of dimensionality.

*Reconstruction-based* methods detect outliers by training models to reconstruct samples from the normal distribution, with high reconstruction errors indicating potential anomalies. Techniques in this category include Hawkins (1974); Hoffmann (2007), which utilize PCA to detect outliers. These methods may fail in situations where the true dependence is more complex than a multivariate normal distribution.

*One-Class-Classification* techniques use discriminatory models to establish a decision boundary that separates normal data from potential anomalies. Successful applications include kernel-based methods, such as Schölkopf et al. (1999); Tax & Duin (2004); Ben-Hur et al. (2001) or tree-based models, such as Liu et al. (2008); Hariri et al. (2021); Guha et al. (2016); Gopalan et al. (2019). Yet these models rely on hyper-parameter tuning and are often constrained in their ability to capture complex or non-linear data distributions effectively (Perera et al., 2021; Seliya et al., 2021).

**MLP.** More recently, MLPs, have gained popularity due to advancements in self-supervised learning and diverse neural architectures. These methods leverage strategies like masking (Thimonier et al., 2024), transformation techniques (Bergman & Hoshen, 2020), and contrastive learning (Qiu et al., 2021; Shenkar & Wolf, 2022), often incorporating attention mechanisms, CNNs, and GANs. Reconstruction loss is another common approach, where autoencoders (Principi et al., 2017; Chen & Konukoglu, 2018; Kim et al., 2020) and GANs (Schlegl et al., 2019) are employed to reconstruct input data and detect anomalies. Additionally, MLPs have been used in one-class classification to isolate inliers from outliers by minimizing the volume of a data-enclosing hypersphere (Ruff et al., 2018), which has been extended in Ruff et al. (2021).

Despite their significant potential, these models face notable limitations. One primary concern is their high-dimensional parameter space, which often exceeds the cardinality of the training set, potentially impeding generalization. Another significant limitation, which can render these models unsuitable for certain applications, is the inherent complexity in interpreting their results. Although methods like those described in Thimonier et al. (2024) provide mechanisms for attribution, these often rely on indirect methods, which may not provide clear insights into model decisions.

## 3 THEORETICAL BACKGROUND

In this section, we provide a concise introduction to copulas and vine copulas, focusing on the aspects relevant to our methodology. For a deeper understanding, we refer the reader to Nelsen (2006); Joe (2014); Czado (2019).

**Copulas.** A copula is defined as a multivariate distribution function $C : [0,1]^d \to [0,1]$ with uniform marginals on $[0,1]$. An important finding in copula theory is due to Sklar (1959):

**Theorem 1 (Sklar's Theorem)** *Given a joint distribution function $F$ with marginals $F_1, F_2, \ldots, F_d$, there exists a copula $C$ such that for all $x_1, x_2, \ldots, x_d \in \mathbb{R}$:*

$$F(x_1, \ldots, x_d) = C\left(F_1(x_1), \ldots, F_d(x_d)\right) \tag{1}$$

*If the marginals $F_i$ are continuous, the copula $C$ is unique.*

*If $F$ is assumed to be absolutely continuous, equation 1 can be differentiated:*

$$f(x_1, \ldots, x_d) = \prod_{i=1}^{d} f_i(x_i) \cdot c(F_1(x_1), \ldots, F_d(x_d)) \tag{2}$$

*where $f_i = \frac{\partial F_i}{\partial x_i}$, $i = 1, \ldots, d$ are the densities of the marginals and $c = \frac{\partial C}{\partial F_1(x_1) \ldots \partial F_d(x_d)}$ denoting the density of the copula.*

Sklar's theorem allows an arbitrary distribution to be decomposed into the one-dimensional marginal distributions $F_i$ and the dependency structure as described by the copula $C$. To estimate a probability distribution $F$ based on a sample, the usual approach is to assume some functional form for the

marginal distributions and the copula. Then the parameters can be determined in sequential fashion: First the marginal distributions are estimated and then the sample is transformed and the parameters of the copula are estimated. The validity of this approach was shown by Joe (2005).

To be flexible in estimating the marginals without making stringent assumptions about their true distribution, we employ a Gaussian kernel density estimator, selecting the bandwidth using the plug-in method described by Sheather & Jones (1991). We then use the kernel estimates to transform the observations to the domain of the copula $[0,1]^d$: $u^{(k)} = (u_1, \ldots, u_d) = (F_1(x_1^{(k)}), \ldots, F_d(x_d^{(k)}))$. Subsequently, we estimate the parameters for a parameterized multivariate copula class using this transformed sample. This can be achieved in a highly flexible way, using vine copulas.

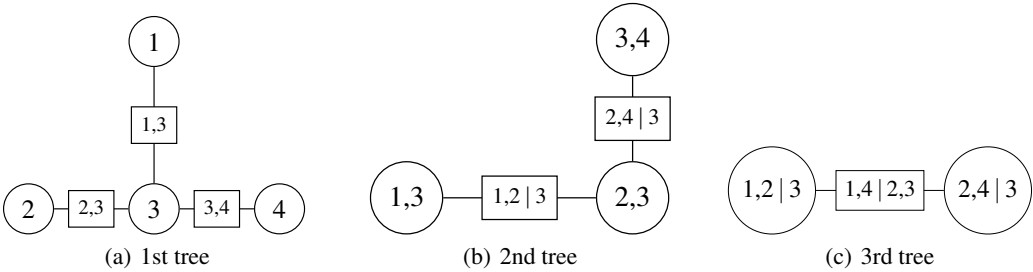

(a) 1st tree            (b) 2nd tree            (c) 3rd tree

Figure 1: Example of a vine copula in four dimensions, consisting of three trees.

**Vine copulas.** This subclass of multivariate copula functions is constructed out of several bivariate copulas, which are arranged in successive levels, the so-called trees (Aas et al., 2009). They can be defined via their density:

$$c(u_1, \ldots, u_d) = \prod_{t=1}^{d-1} \prod_{\{i,j\} \in E_t} c_{i,j;\boldsymbol{d}_{i,j}}(F(u_i \mid \boldsymbol{d}_{i,j}), F(u_j \mid \boldsymbol{d}_{i,j}) \,;\, u_{\boldsymbol{d}_{i,j}}) \tag{3}$$

where $E_t$ denotes the edge set of tree $t$, including tuples of two features which are connected in tree $t$ via bivariate copulas, with respective density $c_{i,j;\boldsymbol{d}_{i,j}}(\cdot, \cdot \,;\, u_{\boldsymbol{d}_{i,j}})$. These copulas are called pair-copulas and they may depend on the point $u_{\boldsymbol{d}_{i,j}}$, where $\boldsymbol{d}_{i,j} \subset \{1, \ldots, d\} \setminus \{i, j\}$ includes the features on which the variables $i$ and $j$ are conditioned. For the first tree this subset is the empty set.

A vine copula can be understood in the following way: The outer product in equation 3 runs over the trees of the vine copula. The trees at each level are spanning trees. For the first tree the nodes are the features and the edges are the pair copulas, connecting two features at a time, see Figure 1(a). The following trees connect adjacent edges of the previous tree, see Figures 1(b),1(c). From the second tree on, the pair-copulas connect not the original features, but the features conditioned on those features $\boldsymbol{d}_{i,j}$, that were bridged with the upstream pair-copulas, see Figures 1(b),1(c). These transformations can be constructed out of the pair-copulas which appear in the previous trees.

Using the greedy algorithm of Dissmann et al. (2013), the vine copula is fitted by determining the structure and pair copulas of the first tree and then processing the remaining trees analogously. The trees at each level are chosen such that they maximize the dependence along their edges, as measured by Kendall's tau. The algorithm captures step by step the most prominent dependencies amongst the features. To accurately model the feature relations, the individual pair-copulas are chosen as kernel-based copulas (Nagler & Czado, 2016; Geenens et al., 2017). To facilitate estimation, equation 3 is simplified by assuming that the pair copulas do not depend on the specific points $u_{\boldsymbol{d}_{i,j}}$. Further the vine is 'truncated' after tree $T \leq d - 1$, meaning that the outer product runs only to $T$. Plugging the thusly modified version of equation 3 into equation 2 yields:

$$f(x_1, \ldots, x_d) = \prod_{i=1}^{d} f_i(x_i) \cdot \prod_{t=1}^{T} \prod_{\{i,j\} \in E_t} c_{i,j}(F_i(x_i \mid \boldsymbol{d}_{i,j}), F_j(x_j \mid \boldsymbol{d}_{i,j})) \tag{4}$$

where $F_{\cdot}(\cdot | \boldsymbol{d}_{\cdot,\cdot})$ concatenates the transformation of the feature to the domain of the copula with the internal transformations of the vine-copula.

## 4 METHOD

We introduce a scalable method that remains sensitive to local deviations from the assumed distribution to detect outliers in high-dimensional data. In this chapter, we provide an intuitive exposition of the outlier score of VC-BOD where, in the first paragraph, we derive the final computation of the outlier score, which consists of the marginal and the dependence scores. We then provide further details on these components in the following paragraphs. In Section A.1 of the Appendix, we provide a more theoretical derivation and further statistical analysis of VC-BOD.

Let $\{\boldsymbol{x}_1, \ldots, \boldsymbol{x}_n\}$ with $\boldsymbol{x}_i \in \mathrm{R}^d$ be a sample of 'normal' observations drawn from an unknown $d$-dimensional random variable $X$ with cumulative distribution function $F_X$. Building on Sklar's Theorem, our approach considers a new observation anomalous if it violates any aspect of the distribution $F_X$. To confirm if this applies, we model the distribution $F_X$ based on the sample, following the approach discussed in Section 3, with Gaussian kernels for the marginals and a truncated vine copula to capture dependence.

Zimek et al. (2012) expose the problems of outlier detection at high dimensions that arise when all features are considered together. We utilize the factorized form of the density in equation 4 and calculate for a new observation an individual score for each constituent of equation 4. Thereby, we transform the possibly high-dimensional problem into several one- and two-dimensional problems. The individual scores are scaled logarithmically, with larger values indicating a higher probability of the new observation being an outlier. For each feature, $f_i$, the marginal score, denoted by $m_i$, takes values in the interval $[0, 1]$. For the pair-copulas $C_{i,j}$, each dependence score, denoted by $d_{ij}$, takes values in the possibly smaller interval $[0, max_{ij}]$, with $max_{ij} \in [0, 1]$. By grouping the scores related to each feature, we obtain our final outlier score:

$$s := \frac{1}{d} \sum_{i=1}^{d} s_i := \frac{1}{d} \sum_{i=1}^{d} \left[ \left(m_i + \sum_{j \in CS_i} d_{i \leftrightarrow j}\right) / \left(1 + \sum_{j \in CS_i} max_{i \leftrightarrow j}\right) \right] \tag{5}$$

where $CS_i$ denotes the copula set of feature $i$, which includes all pair-copulas connecting feature $i$ with other features. The notation $i \leftrightarrow j$ indicates that $i$ is either the first or the second index.

The overall score, which is the average of the feature scores, takes values in $[0, 1]$, with larger values indicating a higher probability that the new observation is an outlier. VC-BOD's algorithmic implementations of training and inference are provided in Appendix A.2, see Algorithms 1 and 2.

**Marginal scores.** For a new observation we calculate for each feature $i = 1, \ldots, d$ a score that quantifies the probability that the $i$-th feature of the observation does not fit with the learned distribution. More specifically, we focus on the probability that a realisation had been observed, which is less, or exactly as, likely as the actual realisation, under the learned distribution. Intuitively, this is the same rationale as with p-values of statistical tests. More formally, this concept utilizes 'minimal volume sets', introduced initially in the works of Einmahl & Mason (1992) and Polonik (1995) and specialized as 'mass-volume curves' for outlier detection by Clémençon & Thomas (2018).

For the new observation $\boldsymbol{y} = (y_1, \ldots, y_d)'$ we first consider the probability of the set $\mathcal{A}_i := \{x \mid f_{X_i}(x) \leq f_{X_i}(y_i)\}$ under the kernel estimator, where $f_{X_i}$ denotes its density:

$$P_{f_{X_i}}(\mathcal{A}_i) = \int_{\mathcal{A}_i} f_{X_i}(x)\, dx \approx \frac{1}{k} \sum_{j=1}^{k} \mathbf{1}_{z_j \leq f_{X_i}(y_i)} =: p_i(y) \tag{6}$$

where the grid-points of $\boldsymbol{z}_i = \{z_1, \ldots, z_k\}$ are constructed as: $z_j := f_{X_i}(F_{X_i}^{-1}(j/(k+1)))$ with $F_{X_i}$ denoting the cdf of the kernel estimator and $k = a^2$ for some $a \in \mathbb{N}$.

Finally the approximated probability is scaled using a logarithm, such that unlikely observations receive an over-proportional score:

$$m_i := -\log_b(p_i(y))$$

The base $b$ of the logarithm is chosen such that $1/k$ is mapped to 0.99. In the case of $p_i(y) = 0$, the score is manually set to 1.

**Dependence scores.**    For the new observation $\boldsymbol{y} = (y_1, \ldots, y_d)'$ we calculate for each pair-copula in equation 4 one outlier score. As in the marginal case, the rationale for each score is drawn from Clémençon & Thomas (2018). To calculate the score $d_{ij}$ corresponding to the copula $C_{i,j}$, we first transform the new point according to the fitted vine copula and calculate the corresponding features: $(u_{y_i}, u_{y_j})$. Next, we approximate the probability of the set $\mathcal{A}_{ij} := \{(u_1, u_2) \,|\, c_{i,j}((u_1, u_2)) \leq c_{i,j}((u_{y_i}, u_{y_j}))\}$ under the copula $C_{i,j}$:

$$
P_{c_{i,j}}(\mathcal{A}_{ij}) = \int_{\mathcal{A}_{ij}} c_{i,j}(u_1, u_2)\, d(u_1, u_2) \approx \frac{1}{k} \sum_{l,l'=1}^{\sqrt{k}} \mathbf{1}_{z_{l,l'} \leq c_{i,j}(u_{y_i}, u_{y_j})} =: p_{ij}(y) \tag{7}
$$

where the elements of the vector $\boldsymbol{z}_{ij} = \{z_{1,1}, \ldots, z_{\sqrt{k}, \sqrt{k}}\}$ are constructed as: $z_{l,l'} := c_{i,j}(l/(\sqrt{k}+1), h_1^{-1}(l'/(\sqrt{k}+1), l/(\sqrt{k}+1)))$, with $h_1^{-1}(\cdot; u_1)$ being the pseudo-inverse of: $h_1(u_2; u_1) = P(U_j \leq u_2 \mid U_i = u_1) = \frac{\partial C_{i,j}(u_1, u_2)}{\partial u_1}$, and the total number of grid-points $k$ being identical to the case of the marginal scores.

We activate the estimated probability with a logarithm, using the same base $b$ as with the marginal scores. For the dependence scores we require an additional multiplicative factor. It is comprised of a component $sod_{ij} \in [0, 1]$, which quantifies the strength of dependence of the corresponding pair-copula $C_{i,j}$. We provide a detailed definition in Appendix A.3. This component is needed because the surface of the density of a pair-copula with only moderate dependence may have no low-density region and thus it would not be possible to confidently declare a single observation an outlier, based on this pair-copula. The second component is the discount factor $\eta^{t-1}$, where $t$ denotes the tree number. This factor compensates for potential violations of the simplifying assumption and accounts for the aggregating estimation errors of copulas in higher trees.

The individual outlier score $d_{ij}$ corresponding to pair-copula $C_{i,j}$ is calculated as:

$$
d_{ij} := max_{ij} \cdot -\log_b(p_{ij}(y)) := sod_{ij} \cdot \eta^{t-1} \cdot -\log_b(p_{ij}(y))
$$

**Influence of truncation.**    An important hyper-parameter of our model is the degree of precision to which the dependency structure is to be approximated. This is controlled via the truncation level $T$ of the vine copula in equation 4. With each additional tree more feature interactions are captured. If the vine copula fit is close to the ground truth, then adding more trees increases accuracy by reducing the number of false negatives, as more aspects of the total dependence are considered, which could potentially be violated. Yet in practice a low level of truncation is advisable: Often, the fit is not perfect and accumulating errors in higher trees can result in the algorithm becoming less accurate. We illustrate this aspect with a theoretical example in Section 6.1. We also provide a statistical analysis concerning this trade-off in the last paragraph of Section A.1 in the Appendix.

**Non-continuous features.**    Some datasets have features that are either discrete or have probability mass significantly greater than zero for some realisations, while the values of a second set are realised only once. The theory described in Section 3 is not directly applicable in those cases. To get around this problem, we take advantage of the helpful property that the outlier score in equation 5 is calculated as the average of the feature outlier scores. We therefore calculate the score, as in equation 5, solely for the continuous features. For the non-continuous features, we only calculate a marginal outlier score in a manner analogous to that of continuous features, with further details provided in Appendix A.4. The final outlier score is determined as the average of all feature scores.

**Complexity Analysis.**    Let $d$ be the number of features, $n$ the sample size, $k$ the grid size, and $T$ the truncation level. The complexity to calculate an outlier score for a new point is $O(f_1(k) \cdot d \cdot (T+1))$, where $f_1(k) = O(k)$. The training complexity comprises fitting the kernel estimator and the vine copula as well as calculation of the scores of the training sample, leading to a training complexity of: $O((f_2(n) + f_3(n,k)) \cdot d \cdot (T+1))$ with $f_2(n) = O(n)$ and $f_3(n,k) = O((n+k) \cdot \log(k))$. For detailed information about training and inference times on different datasets, we refer to the Appendix D.2.

## 5 EXPLAINABILITY AND FEATURE ATTRIBUTION

One of the main advantages of our approach is its high level of explainability. VC-BOD offers the following procedure to analyse a new observation:

1. Decision if a new observation is to be classified as an outlier, based on a comprehensible criterion.

2. If an observation is classified as an outlier: Identification of those features that are decisive for this classification.

3. For each decisive feature: Distinction whether it was flagged because it does not fit with the learned marginal distribution or because of the way it interacts with other features. (Only for truncation level $T \geq 1$)

VC-BOD is trained based on a sample of 'normal' observations and a confidence level $\tau \in (0; 1)$, e.g. $\tau = 0.95$, is set for the analysis.

For each new observation $obs$, the outlier score $s(obs)$ is calculated according to equation 5. Next, determine the quantile $q(obs)$ of this score compared to those of the training sample, i.e. determine the fraction of the training scores smaller than $s(obs)$. If $q(obs)$ exceeds $\tau$, then $obs$ is classified as an outlier.

If the observation is identified as an outlier, it can be assessed for each feature $i \in \{1, \ldots, d\}$ whether it plays a significant role in the classification. To do so, determine the quantile $q_i(obs)$ of the feature score $s_i(obs)$ from equation 5 amongst the $i$-th feature scores of the sample. If $q_i(obs) > \tau$, feature $i$ contributes significantly to the observation being classified as an outlier. If no feature crosses the quantile threshold, then the observation is anomalous because of the aggregated influence of all features.

For each of the significant features it can be decided why it was flagged. Determine the quantile $q_{m_i}(obs)$ of the $i$-th marginal outlier score $m_i$ amongst the corresponding values of the training sample. If $q_{m_i}(obs) > \tau$, then the $i$-th feature of $obs$ does not fit with the learned marginal distributions. Else, if $q_{m_i}(obs) \leq \tau$, the feature is considered anomalous as it violates both the marginal distribution and the established dependence with other features. However, since the marginal violation alone did not trigger the quantile threshold, the violation of the learned dependence structure can be regarded as significant.

This level of feature attribution redefines the current standard in outlier detection, offering valuable insights that can greatly benefit a wide range of practical applications. In Appendix B.1, we validate our attribution framework through a theoretical experiment. We also present several plots in Appendix B.2 that provide a clear understanding of why observations are labeled as outliers. In Section B.3, we demonstrate the attribution framework using the 'wine' dataset, which is also part of the empirical study presented in the second part of Section 6.

## 6 EXPERIMENTS

We first examine the impact of the truncation level on VC-BOD's detection capabilities, followed by an extensive empirical study to evaluate its performance.

### 6.1 INFLUENCE OF TRUNCATION

To illustrate how the truncation level can influence the resulting outlier detection model, consider the distribution displayed in Figure 2. This distribution is detailed in Appendix C.1. If the truncation level is chosen as $T = 0$, no vine copula is fitted and only the marginal distributions are regarded. For $T = 1$ the first tree is fitted, which, in the case displayed in Figure 2, connects features one and two (2(a)) and features two and three (2(b)), since the dependence between those pairs is more pronounced than between features one and three (2(c)). For $T = 2$ the whole vine is fitted and, in addition to the first tree, the second tree connects features one and three, given feature two. This conditional distribution does not correspond exactly to the unconditional situation displayed in 2(c), but is very similar. We train the outlier detector in each case based on the same sample. In the first

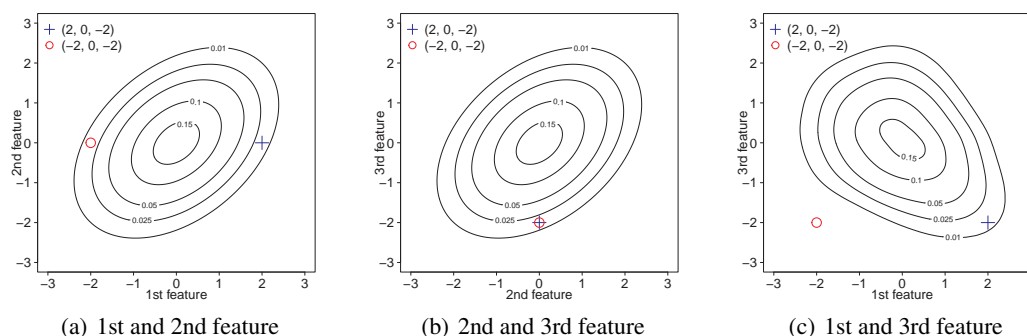

(a) 1st and 2nd feature      (b) 2nd and 3rd feature      (c) 1st and 3rd feature

Figure 2: Pair-density plots of a three-dimensional distribution where the first and the second feature 2(a), as well as the second and third feature 2(b), are positively correlated, but the first and the third feature 2(c) are negatively correlated. All three marginal distributions follow the standard normal distribution. The blue point ($+$) is in-distribution, the red point ($\circ$) is out-of-distribution.

two cases it is not possible to separate the points, as for $T = 0$ and $T = 1$ the red point ($\circ$) and the blue point ($+$) receive almost identical scores. For $T = 2$ the score of the blue point ($+$) is $s = 0.18$, which is in the 93$^{\text{rd}}$ percentile compared to the scores of the training data. The score for the red point ($\circ$) is $s = 0.36$, which exceeds all training scores, falling into the 100$^{\text{th}}$ percentile. Only by considering the full dependency structure, the points can be effectively separated and correctly classified.

## 6.2 EMPIRICAL EVALUATION

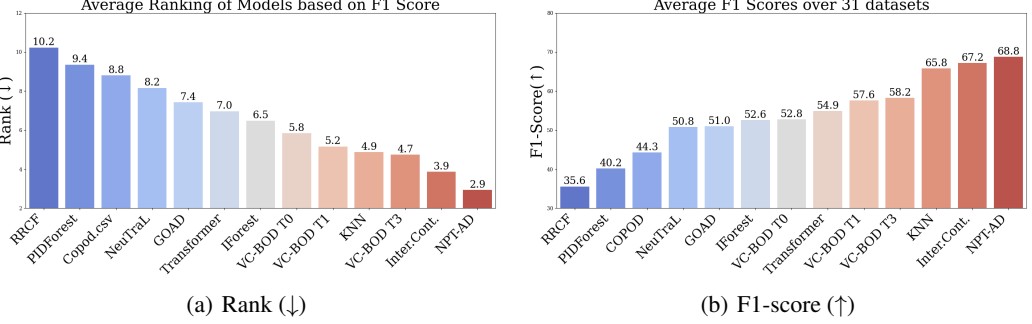

(a) Rank ($\downarrow$)                  (b) F1-score ($\uparrow$)

Figure 3: **Benchmarking VC-BOD**. We evaluate three variants of VC-BOD at truncation levels 0, 1, and 3, referred to as T0, T1, and T3, respectively. Following Thimonier et al. (2024), average performance over 40 seeds across 31 data-sets was computed. Figure 3(b) illustrates the average F1 score, where a higher average score is better. Figure 3(a) displays the average rank based on the mean F1-score, where a lower average rank is better. Detailed metrics are provided in the Appendix D, see Table 4 and Table 5.

We evaluate three variants of VC-BOD: VC-BOD T0, where we only consider the marginals. VC-BOD T1, where the truncation level is set to one and VC-BOD T3, where the truncation level is set to three. We conduct an empirical evaluation following the well-established experimental setup used in Shenkar & Wolf (2022); Thimonier et al. (2024); Bergman & Hoshen (2020); Zong et al. (2018), which can be stated as follows: Denote by $\mathcal{N}$, $\mathcal{A}$ the sets of normal and anomalous observations. Training and test set are chosen as: $\mathcal{N}_{\text{train}} \subset \mathcal{N}$, with $|\mathcal{N}_{\text{train}}| = |\mathcal{N}|/2$ and $\mathcal{V}_{\text{test}} = (\mathcal{N} \backslash \mathcal{N}_{\text{train}} \cup \mathcal{A})$. For each observation of the test set an outlier score is calculated using the VC-BOD variants T0, T1 and T3, each trained on $\mathcal{N}_{\text{train}}$. Observations are flagged as outliers if their score surpasses a threshold, which is set such that the total number of flagged anomalies matches the cardinality of $\mathcal{A}$.

Following Shenkar & Wolf (2022); Thimonier et al. (2024), we benchmark our model against five widely-used non-deep learning models. Specifically, we include KNN (Ramaswamy et al., 2000) from the distance-based family, IForest (Liu et al., 2008) and PIDForest (Hariri et al., 2021; Guha et al., 2016) from the family of discriminatory models, and COPOD (Li et al., 2020), from the family of distribution based methods. Additionally, we compare our model against several deep learning models, including GOAD (Bergman & Hoshen, 2020), NeuTraL-AD (Qiu et al., 2021), the contrastive approach InterCont. proposed by Shenkar & Wolf (2022), and the transformer-based models Transformer (Vaswani et al., 2017) and NPT-AD (Thimonier et al., 2024). We excluded DROCC (Goyal et al., 2020) from the set of outlier detection methods because it achieves subpar performance and fails to yield any results in approximately one-third of the considered datasets.

We evaluate performance using the F1 score and the average rank. Additional results, based on the AUC, are provided in Table 6 in Appendix D. All referenced metrics were obtained from Thimonier et al. (2024).

**Datasets.** We use a similar set of benchmark datasets as Shenkar & Wolf (2022); Thimonier et al. (2024). In total 31 datasets are considered, including 28 multi-dimensional point datasets sourced from the "Outlier Detection DataSets" (ODDS) repository[1]. Following Thimonier et al. (2024) the datasets 'Heart' and 'Yeast' are excluded and the two commonly-used datasets 'Arrhythmia' and 'Thyroid', as well as the three real-world datasets: 'Fraud', 'Campaign', and 'Backdoor', obtained from Han et al. (2022) are included. We provide a detailed overview of the dataset characteristics in Table 2, see Appendix C.2. Due to hardware constraints VC-BOD was trained using only 20% of the available training data for the datasets 'Fraud' and 'Mullcross'.

**Results.** We begin by presenting initial observations based on the average F1 score and the average rank to provide an overview of the model performances. In the second step, we employ a statistical significance test to assess whether the observed differences in the average rank between models are statistically meaningful.

As shown in Figure 3(a), VC-BOD T3 achieves lowest average rank among all classical models and third lowest average rank among all models, surpassed only by the latest transformer-based models InterCont. and NPT-AD. This shows the notable generalization capabilities of VC-BOD T3, demonstrating its effectiveness even relative to the best MLP-based models. This achievement is significant, especially since our method provides a closed-form interpretation of results, unlike MLPs, which suffer from issues concerning explainability and feature attribution.

Figure 3(b) also substantiates our theoretical insights from Section 6.1, that increasing tree depth enhances performance and generalization across various datasets. This is evidenced by an improvement of the F1 score from 57.6 to 58.2 and an improvement in overall mean rank from 5.2 to 4.7. This is further highlighted in Table 3 in Appendix D. When evaluating the average F1 score, displayed in Figure 3(b), VC-BOD T3 and VC-BOD T1 rank as state of the art methods, trailing only behind KNN, InterCont. and NPT-AD. A detailed overview of all metrics is provided in the Appendix D, see Table 4 and Table 5.

Notably, even the marginal-only variant VC-BOD T0 displays competitive performance with a mean F1-score of 52.8 and a better average rank than the well-established classical method IForest and the MLP models GOAD, NeuTraL-AD.

To validate the ranking of the evaluated methods, as illustrated in Figure 3(a), we conducted a Friedman-Nemenyi test to determine statistically significant differences, as detailed in Campos et al. (2016); Liu et al. (2019). Appendix D.1 presents the results, showing that VC-BOD outperforms several baseline models, including COPOD, PIDForest, RRCF, and NeuTraL. Additionally, our analysis indicates no statistically significant difference between VC-BOD and KNN, nor between VC-BOD and leading deep learning approaches such as NPT-AD. These findings position VC-BOD at the forefront of performance, complemented by its provision of one of the most advanced explainability frameworks available.

---

[1] http://odds.cs.stonybrook.edu/

# 7 DISCUSSION

The main limitations of VC-BOD are due to its usage of vine copulas to capture dependence. In situations with high feature dimension and simultaneously high sample size, fitting the vine copula leads to long run-times for model training. Inference remains fast, and the margin-only approach does not suffer from these complexity issues. Notably, all vine copula computations currently use multi-threaded C++ on CPU cores, and training time could likely be reduced with GPU utilization.

By laying the foundation for highly explainable outlier detection, our work invites future research to adapt our methodology to unsupervised outlier detection. This requires adjustments since VC-BOD relies on kernel methods which reflect the training data very accurately. While effective in semi-supervised settings, where the model is trained on uncontaminated data, this can become more problematic in unsupervised settings, as the proportion of outliers in the training data increases.

In this work, we introduced Vine Copula-Based Outlier Detection (VC-BOD), significantly advancing the field of outlier detection by providing enhanced explainability. Our empirical evaluations demonstrate that VC-BOD achieves state-of-the-art detection capabilities. By incorporating both marginal and dependence-related information into a closed-form equation, VC-BOD offers interpretability in complex, potentially high-dimensional settings - a crucial advantage in domains requiring transparent decision-making. In conclusion, VC-BOD demonstrates the continuing value of probabilistic approaches in outlier detection, providing a powerful and highly interpretable alternative to deep-learning based models.

## REPRODUCIBILITY STATEMENT

We ensure the reproducibility of our work through the use of a deterministic algorithm, the code for which will be made publicly available. This code is designed to run efficiently on standard laptops without the need for GPUs. Additionally, our empirical evaluations are conducted using exclusively datasets that are publicly accessible. This approach guarantees that our findings can be independently verified and replicated.

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

STRUCTURE OF THE APPENDIX

The structure of the appendix is based on the order of the chapters in the main text. In Section A we give a statistical view on VC-BOD. We further provide an algorithmic implementation of VC-BOD. In Section B we focus on the explainability of VC-BOD. We go through the procedure outlined in the paper using a synthetic dataset as an example and also present diagnostic plots that allow for a clear understanding of why observations are classified as outliers. In Section C, we first provide more information about the synthetic and empirical datasets used in our experiments, followed by Section D, where we supply additional material regarding the empirical evaluation of VC-BOD.

# A  METHOD

This section offers a comprehensive overview of VC-BOD. In the first Subsection, we provide a theoretical justification for VC-BOD by framing it within the context of statistical hypothesis testing. In Subsection A.2, we provide a detailed overview of the algorithmic implementation of VC-BOD. In Subsection A.3, we define the strength-of-dependence parameter $sod_{ij}$, which was introduced in the main text. Subsection A.4 details calculating the marginal scores $m_i$ for non-continuous features.

## A.1  STATISTICAL VIEW OF VC-BOD

In this section, we analyze VC-BOD from a statistical standpoint. First, we explicitly outline the assumptions underlying the method. Next, we conceptualize VC-BOD as a statistical test, establishing its theoretical framework. Finally, we explore the trade-off between accuracy and the increasing complexity introduced by a growing truncation level $T$.

**Assumptions.**   In the context of VC-BOD, outlier detection is approached as an out-of-distribution detection task, a common strategy in distribution-based outlier detection. This method aims to estimate the true underlying distribution based on a given sample and then determine whether a new data point is unlikely under the inferred distribution. The following assumptions are made:

(A1)  The training sample consists of i.i.d. realizations from an unknown distribution.

(A2)  The unknown distribution allows for a vine copula specification, which can be determined using Dissmann's algorithm with Gaussian kernels for the marginals and Gaussian kernel copulas for the pair-copulas, where:

    (A2-1)  The marginal distributions are correctly specified.

    (A2-2)  The pair-copulas in the trees $T_i$ with $i = 2, \ldots, T$ are independent of the conditioning variables.

    (A2-3)  The pair-copulas in the trees $T_i$ with $i = 1, \ldots, T$ are correctly specified.

Assumption (A1) ensures that the underlying distribution, representing the data-generating process, can be effectively learned from the sample. The additional assumptions, grouped under (A2), pertain to the distributional properties of the underlying distribution. Since VC-BOD employs kernel estimators for both the marginal distributions and the pair copulas of the vine copula, explicitly detailing all assumptions becomes cumbersome. Specifically, (A2-1) assumes that each marginal distribution can be expressed as a mixture of (truncated) Gaussian kernels. Meanwhile, (A2-2) represents the simplifying assumption that all pair copulas from tree $T_2$ onward are independent of the conditioning variable(s), meaning:

$$c_{i,j;\boldsymbol{d}_{i,j}}(F(u_i \mid \boldsymbol{d}_{i,j}), F(u_j \mid \boldsymbol{d}_{i,j}) \, ; \, u_{\boldsymbol{d}_{i,j}}) = c_{i,j}(F(u_i \mid \boldsymbol{d}_{i,j}), F(u_j \mid \boldsymbol{d}_{i,j})) \quad \forall (u_i, u_j) \in [0,1]^2$$

. (A2-3) assumes that all pair-copulas $C_{i,j}$ are mixtures of a number of Gaussian kernels, i.e. their density is of the following form:

$$c_{i,j}(u,v) = \frac{1}{n} \sum_{k=1}^{n} \frac{N\big(\Phi^{-1}(u), \Phi^{-1}(v) \mid \Phi^{-1}(u_k), \Phi^{-1}(v_k), \Sigma\big)}{\phi\big(\Phi^{-1}(u)\big)\phi\big(\Phi^{-1}(v)\big)}$$

where $N(\cdot, \cdot \mid \nu_1, \nu_2, \Sigma)$ denotes a bivariate Gaussian density with mean vector $(\nu_1, \nu_2)$ and co-variance matrix $\Sigma = n^{-1/3} \mathrm{Cor}(\Phi^{-1}(U), \Phi^{-1}(V))$. Further $\phi$ and $\Phi^{-1}$ respectively denote the density and quantile function of the standard normal distribution, see Nagler & Czado (2016) for more details.

**VC-BOD interpreted as a statistical test.** VC-BOD, as an outlier-detection method to evaluate a single new observation in the context of a given sample, can be interpreted as a statistical test. Since an outlier is defined as a point that lies outside the distribution, the null hypothesis asserts that the new observation originates from the same data-generating process as the training data sample. Conversely, the alternative hypothesis posits that the observation is out-of-distribution and thus an outlier. Accordingly, the hypothesis pair underlying VC-BOD is defined as follows:

$$H_0 : \text{new observation stems from underlying distribution}$$
$$H_1 : \neg H_0 : \text{new observation is an outlier}$$

Using a truncated vine copula (see equation 4 in the main text), we gain access to all univariate marginal distributions as well as specific bivariate (conditional) components of the overall distribution, represented by the pair copulas of the vine copula. Let us assume there are $N$ univariate and two-dimensional distributions in total.

Under the above null hypothesis, it follows that the new observation matches with all lower-dimensional aspects of the whole distribution:

$$H_0 \longrightarrow \{H_0^{(1)}, \ldots, H_0^{(N)}\}$$

where $H_0^{(k)}$ is the null hypothesis that the new observation matches the $k$-th. aspect of the whole distribution, which is either a marginal or a (conditional) bivariate distribution, linking two variables (conditional on other variables).

If there is sufficient evidence that the individual $H_0^{(k)}$'s on the right-hand side of the implication above are not simultaneously true, then the global $H_0$ can be rejected, and the new observation is declared an outlier. Thus, VC-BOD can be viewed as a multiple testing procedure, where the joint validity of the individual null hypotheses $H_0^{(k)}$ serves as a surrogate for the global $H_0$ that the new observation is not an outlier.

To test each individual null hypothesis $H_0^{(k)}$, we calculate the corresponding quantity $p_i(y)$ or $p_{ij}(y)$ as described in the main text. Given the assumptions and under $H_0^{(k)}$, the quantity is uniformly distributed on $[0, 1]$ and represents a p-value $p_k$ for $H_0^{(k)}$.

To motivate the general functional form of the outlier score in VC-BOD, we begin by focusing on the aggregation of individual p-values $p_1, \ldots, p_N$ into a single meta-statistic. We draw inspiration from Fisher's combined probability test, a well-established method for aggregating multiple p-values into a unified measure of significance:

$$s_F = 2 \cdot \sum_{i=1}^{N} -1 \cdot \log(p_i)$$

If the individual p-values are independent, then, given that the individual null hypotheses are simultaneously true, $s_F$ follows a chi-square distribution with $df = 2 \cdot N$, see Littell & Folks (1971).

For our task of explainable outlier detection we scale the p-values corresponding to pair-copulas and group p-values related to the same variable to obtain the form of the outlier score presented in the main text in equation 5. This transformation complicates the distribution of the test statistic. A more severe problem, however, is that assuming the individual p-values to be independent is not reasonable. As a result, the distribution under $H_0$ is not known. We, therefore, work with the empirical distribution, which is obtained by calculating the values of the test statistic of the sample. Given the assumptions (A1) and (A2), the empirical distribution can be used as a reliable estimate for the unknown true distribution.

The criterion for identifying an outlier, presented in the first step of the procedure laid out in Section 5 of the main text, is, given the assumptions (A1) and (A2), an empirical approximation of the p-value to test a surrogate version of the global $H_0$. If the p-value is deemed too small, the global $H_0$

is rejected, and the observation is declared an outlier. However, if the p-value is above the critical threshold, the observation might still be an outlier; see the discussion in the following paragraph.

**Trade-off between accuracy and growing set of assumptions.** If it is established that a new observation violates an aspect of the overall distribution, then this is a sufficient condition that the observation is out-of-distribution also in the context of the whole distribution. An observation may violate an aspect captured in a later tree of the vine copula while matching every aspect captured in the lower trees. An example of this is provided in Section 6.1 of the main text.

Therefore, adding more trees, i.e., working with a larger truncation level $T$, can improve accuracy by reducing the number of false negatives, i.e., anomalous observations that are not identified as outliers. This is because more sufficient conditions for the observation to be an outlier are checked.

However, by increasing the truncation level $T$, the assumptions made in (A2) become more restrictive. Especially the simplifying assumption (A2-2) is often inappropriate in practice (Acar et al., 2012; Spanhel & Kurz, 2019). Note that the variants VC-BOD T0 and VC-BOD T1, where the truncation level is set to $T = 0$ and $T = 1$ respectively, do not require (A2-2). Another problem that comes into play when considering more aspects of the total distribution is that the combined test statistic loses power; see Shaffer (1995).

To achieve a good balance, we recommend using a relatively low level of truncation. The vine copula is fitted iteratively, identifying the (conditional) bivariate distributions with the strongest dependencies at each step, indicating that they are well-suited for out-of-distribution detection. For this reason, the truncated vine copula serves as a useful tool for identifying those feature mappings that are most appropriate for outlier detection. In the context of VC-BOD, the truncated vine copula should therefore be regarded primarily as a tool for feature extraction, rather than as an approximation to the entire distribution.

### A.2 ALGORITHMIC IMPLEMENTATION OF VC-BOD

Here we provide an algorithmic description of the training, see Algorithm 1, and inference phase, see Algorithm 2, of VC-BOD.

Algorithm 1: In the first stage, we address the feature distributions by distinguishing between continuous and non-continuous features. In both cases, we store information about the feature distributions to compute outlier scores later. In the second stage, we fit a vine copula to the transformed continuous features. For each of its pair-copulas, we calculate the factor $\max_{ij}$ and, if the pair-copula does not exhibit significant dependence, we exclude it from further considerations. Finally, we calculate the scores for the training sample.

References to detailed explanations in the main text correspond to the first three subsections of Section 4. The calculations of the density grids (lines 6 and 25) and individual outlier scores (lines 7 and 26), as well as the final outlier scores (line 32), are thoroughly explained there. The handling of non-continuous features (lines 10–14) is discussed in Appendix A.4. The definition of the strength-of-dependence parameter $sod_{ij}$ (line 22) is provided in Appendix A.3.

Algorithm 2: The outputs from the first algorithmic stage are used as input alongside the new observations. The algorithm iterates over the test data, computing the marginal scores and dependence scores for each test instance. Based on these individual scores, the final outlier score for each instance is calculated according to equation 5 in the main text. By comparing these scores against the $\tau$-quantile of the training scores, we assign a label to each test instance, where '1' indicates that the observation is declared an outlier. The calculations of the individual outlier scores (lines 5 and 12) and the final outlier score (line 14) are explained in detail in the main text.

We make the following choices for the involved hyper-parameters. The number of total grid-points, both for the marginal scores and the dependence scores, is chosen as $k = 10\,000$. The discount factor, which appears in the calculation of the dependence scores, is chosen as $\eta = 0.75$.

### A.3 STRENGTH-OF-DEPENDENCE PARAMETER

In this subsection we give more details on the 'strength-of-dependence' parameter $sod_{ij}$ which is calculated for each pair-copula $C_{i,j}$ to quantify the amount of dependence it captures. While there

---

**Algorithm 1** VC-BOD - Train

---

**Require:** Multivariate data-sample $\boldsymbol{X} \in \mathbb{R}^{n \times d}$, truncation level $T \leq d - 1$, discount factor $\eta$
 1: Initialize empty list *marg_info*         ▷ Container for marginal information
 2: **for** $i = 1$ to $d$ **do**         ▷ Loop over features
 3:      Let $\boldsymbol{x}_i = \{X_{1i}, \ldots, X_{ni}\}$ be the realisations of feature $i$
 4:      **if** feature $i$ is continuous **then**         ▷ Continuous feature
 5:          Fit kernel density estimator $KDE_i$ to $X_i$
 6:          Compute grid $\boldsymbol{z}_i$
 7:          Calculate marginal training scores: $\boldsymbol{m}_i = \{m_i\}_1^n$
 8:          Store $KDE_i$, $\boldsymbol{z}_i$ and $\boldsymbol{m}_i$ in *marg_info[i]*
 9:      **else**         ▷ Non-continuous feature
10:          Identify points $\boldsymbol{e}_i$ in $\boldsymbol{x}_i$ that receive substantial empirical probability mass
11:          Compute their respective probabilities $\boldsymbol{p}_i$
12:          Fit kernel density estimator $KDE_i$ to the set non-discrete points (if non-empty)
13:          Compute grid $\boldsymbol{z}_i$ for non-discrete points (if non-empty)
14:          Calculate marginal training scores: $\boldsymbol{m}_i = \{m_i\}_1^n$
15:          Store $\boldsymbol{e}_i$ and $\boldsymbol{p}_i$, $KDE_i$ and $\boldsymbol{z}_i$ (if defined), $\boldsymbol{m}_i$ in *marg_info[i]*
16:      **end if**
17: **end for**
18: Transform continuous features to domain of copula: $\boldsymbol{U}_c = \left(F_{KDE_{c_1}}(\boldsymbol{x}_{c_1}), \ldots, F_{KDE_{c_d}}(\boldsymbol{x}_{c_d})\right)$
19: Fit vine copula $vc$ with truncation level $T$ to $\boldsymbol{U}_c$
20: Initialize empty list *dep_info*         ▷ Container for dependence information
21: **for** $C_{ij}$ in {pair-copulas of $vc$} **do**         ▷ Loop over pair-copulas
22:      Calculate $sod_{ij}$-parameter for copula $C_{ij}$
23:      Calculate $\max_{ij} = sod_{ij} \cdot \eta^{tree(C_{ij})-1}$
24:      **if** $\max_{ij} > 0.01$ **then**         ▷ Skip insignificant pair-copulas
25:          Compute grid $\boldsymbol{z}_{ij}$
26:          Calculate dependence training scores: $\boldsymbol{d}_{ij} = \{d_{ij}\}_1^n$
27:      **end if**
28:      Store $\max_{ij}$ and $\boldsymbol{z}_{ij}$, $\boldsymbol{d}_{ij}$ (if defined) in *dep_info[i, j]*
29: **end for**
30: Initialize empty vector *train_scores*         ▷ Container for scores of training sample
31: **for** $i = 1$ to $n$ **do**
32:      $train\_scores[i] = s(i)$         ▷ Calculate outlier score of training observation $i$
33: **end for**
     **return** [*marg_info*, *dep_info*, *train_scores*]         ▷ Return relevant info for VC-BOD - Test

---

---

**Algorithm 2** VC-BOD - Test

---

**Require:** Multivariate test data-sample $\boldsymbol{X} \in \mathbb{R}^{n' \times d}$, threshold $\tau$, output of VC-BOD - Train
1: Determine $\tau$-quantile $q_\tau$ of outlier scores of the training data
2: $scores, labels = (0, \ldots, 0) \in \mathrm{R}^{n'}$           ▷ Initialize vectors containing scores and labels
3: **for** $k = 1$ to $n'$ **do**               ▷ Loop over new observations
4:     **for** $i = 1$ to $d$ **do**                  ▷ Loop over features
5:        Calculate marginal outlier score $m_i^k$ of observation $k$
6:     **end for**
7:     $\boldsymbol{u}_k = (F_{KDE_{c_1}}(u_{c_1}^k), \ldots, F_{KDE_{c_d}}(u_{c_d}^k))$     ▷ Transform continuous features of observation $k$
8:     **for** $C_{ij}$ in {pair-copulas of $vc$} **do**             ▷ Loop over pair-copulas
9:        **if** $\max_{ij} < 0.01$ **then**
10:           continue             ▷ Skip insignificant pair-copulas
11:        **end if**
12:        Calculate dependence outlier score $d_{ij}^k$ of observation $k$
13:     **end for**
14:     $scores[k] = s(k)$             ▷ Calculate outlier score of test observation $k$
15:     **if** $s(k) > q_\tau$ **then**             ▷ Assign labels based on threshold
16:        $labels[k] = 1$
17:     **end if**
18: **end for**
     **return** $[scores, labels]$             ▷ Return final outlier scores and labels

---

already exist several concepts, like the Pearson correlation, Spearman's rho, Kendall's tau or Ho-effding's D, we found that none of them were ideal for the purpose of outlier detection. This is because with outlier detection a single new observation has to be evaluated in the context of the copula. Most relevant for this task is the existence of low-density regions, as a new observation is likely to be an outlier, if it falls into such a region. Therefore we derive a simple criterion, which aims to evaluate the maximal height difference of the bivariate copula density. The rationale is as follows: If the copula in question is the independence copula, where it is impossible to detect an anomaly, than the density is constant and equal to one everywhere and the maximal height difference is zero. If, on the other hand, there are low-density regions, then there must also exist a region at an average height greater than one, as the density over the whole unit hyper-cube integrates to one. The height difference is then greater than zero. For sufficiently smooth density surfaces, which are given with appropriately specified kernel copulas, the following holds. The more wide-reaching and 'lower' the low-density region, the more wide-reaching and 'higher' the density of another region and the larger the height difference. We therefore use the maximal height difference as a proxy for the copula's suitability to detect outliers.

To calculate the maximal height difference of the kernel copula $C_{i,j}$, evaluate its density at the grid points $z_{kl} = (z_k, z_l)$, where $\{z_1, \ldots, z_n\}$ is an equidistant grid on $[0, 1]$. The resulting density values are sorted in ascending fashion: $\{d_1^{ij}, \ldots, d_{n^2}^{ij}\}$. Next, we calculate:

$$max\text{-}diff_{ij} := p \cdot \Big( \sum_{k=\lfloor (1-p) \cdot n^2 \rfloor}^{n^2} d_k^{ij} - \sum_{k=1}^{\lceil p \cdot n^2 \rceil} d_k^{ij} \Big)$$

The strength-of-dependence parameter $sod_{ij}$ corresponding to copula $C_{i,j}$ is defined as:

$$sod_{ij} := \min \Big( \max \Big( \frac{max\text{-}diff_{ij} - diff_{\min}}{diff_{\max} - diff_{\min}}, 0 \Big), 1 \Big)$$

The strength-of-dependence is therefore equal to zero for $max\text{-}diff_{ij} \leq diff_{\min}$, equal to one for $max\text{-}diff_{ij} \geq diff_{\max}$ and scales linearly in-between.

We made the following choices: $n = 100$, $p = 0.05$, $diff_{\min} = 1$, $diff_{\max} = 4$.

A.4   NON-CONTINUOUS FEATURES

Here, we discuss the calculation of the marginal outlier scores $m_i$ of the non-continuous features.

To identify non-continuous features, we set a threshold for the probability mass of a single realisation. If it is exceeded by at least one realisation, we say that the corresponding feature is non-continuous. We used a threshold of $\tau = 2/3$. Using kernel copulas as pair copulas in the vine copula construction allows us to select this comparatively large value.

For non-continuous features, we differentiate two cases: Features that are discrete and features that have probability mass significantly greater than zero for some realisations, while the values of a second set are realized only once.

For the first case, we calculate $p_i$ as:

$$p_i := 1 - \Big( \sum_{x_j : f(x_j) > f(y)} f(x_j) \Big)$$

where $f$ denotes the empirical probability function of the training sample.

For the second case, we determine the vector $z_i$ for the continuous realisations and the empirical distribution function $f_d$ for the discrete realisations and calculate:

$$p_i(y) = \begin{cases} 1 - \big( \sum_{x_j \text{ discrete and } f_d(x_j) > f_d(y)} f_d(x_j) \big) & \text{if } y \text{ in discrete set,} \\ \big(1 - \sum_{x_j \text{ discrete}} f_d(x_j)\big) \cdot \frac{1}{k} \sum_{j=1}^{k} \mathbf{1}_{f_{X_i}(z_j) \leq f_{X_i}(y_i)} & \text{else.} \end{cases}$$

In both cases, the marginal score of feature $i$ is defined as:

$$m_i := -\log_b(p_i)$$

where the basis $b$ is chosen such that a realisation that was observed only once in training receives a score of $0.99$. If $p_i = 0$, then $m_i$ is manually set to one.

## B  EXPLAINABILITY AND FEATURE ATTRIBUTION

### B.1  VALIDATION OF THE FEATURE ATTRIBUTION FRAMEWORK

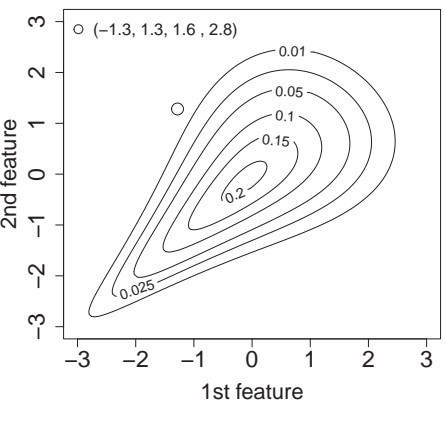
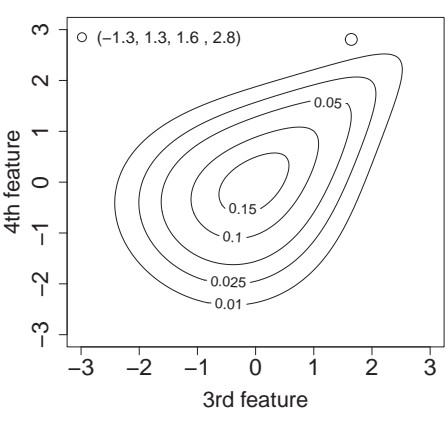

(a) 1st and 2nd feature                    (b) 3rd and 4th feature

Figure 4: Pair-density plots of a four-dimensional distribution where the first and the second feature 4(a) are related such that simultaneous small realisations are likely and the third and fourth feature 4(b), are related such that simultaneous large realisations are likely. All four marginal distributions follow the standard normal distribution. The indicated point constitutes an outlier. The features responsible for this classification are features 1, 2 and 4. Features 1 and 2 break with the dependency structure, while feature 4 breaks with its marginal distribution.

In this section, we expand on our explainability framework of Section 5 of the main text. We will do so by reference to a specific four-dimensional distribution which is depicted in Figure 4. This distribution is characterized by a block-dependence structure, where features 1 and 2 and features 3 and 4 depend on one another, but there is no dependence between the two blocks. The features of the first block are defined via a Clayton copula with parameter $\theta = 1.5$. The features of the second block are defined via a 180-degree rotated Clayton copula with parameter $\theta = 0.75$. All marginal distributions follow the standard normal distribution. In Figure 4 there is one point indicated, which is defined as $(qn(0.1), qn(0.9), qn(0.95), qn(0.9975))' = (-1.3, 1.3, 1.6, 2.8)'$, where 'qn' denotes the quantile function of the standard normal distribution, which is the inverse of its cumulative distribution function.

We will now go through the steps laid out in Section 5 to analyse the indicated point in the context of the distribution displayed in Figure 4:

For the first step, we run Algorithm 1, to a sample of the distribution and choose $T = 3$ as the truncation level. Next, we use Algorithm 2, to evaluate the indicated point. For our analysis, we set the confidence level to $\tau = 0.95$. The score of the indicated point is in the $99.9^{\text{th}}$ percentile of the training scores. Thus the point is identified as an outlier.

Since the indicated point constitutes an outlier, we continue with the second step, where we identify those features that strongly contribute to this classification. To achieve this, we determine the quantile of each feature score, $s_i$ from equation 5, relative to the corresponding feature scores in the training set. VC-BOD reveals that the quantiles for features 1-4 are: $0.99, 0.99, 0.83, 0.99$. Therefore, features 1,2 and 4 are to be regarded as anomalies, while feature 3 is inconspicuous.

In the third step, we determine if each significant feature breaks with its own univariate marginal distribution or if it is anomalous because of the way it interacts with other features. For this, we

determine the quantile of each marginal feature score, $m_i$'s from equation 5, relative to the corresponding marginal feature scores in the training set. VC-BOD reveals that the quantiles for features 1, 2, and 4 of the indicated point are: $0.80, 0.80$ and $0.99$. We can conclude that the first two features break with the dependency structure, as $0.8 < 0.95$, and the fourth feature breaks with its marginal distribution, as $0.99 > 0.95$.

Note that the marginal quantiles of the scores are in line with the definition of the indicated point. For example, the 1st and 2nd features of the point are respectively chosen to be in the 10th percentile and 90th percentile of the standard normal distribution. As this distribution is uni-modal and symmetric around its mean, this means that $20\%$ of all possible realisations are less likely than the observed features. This circumstance is reflected in the quantiles of the marginal outlier scores being $0.8$ in both cases.

Further, we can infer from Figure 4 that the attribution of the significant features is correct: Features 1 and 2 of the distribution are related such that simultaneous small realisations are likely, see Figure 4(a), but the 1st feature of the indicated point is small, while its 2nd feature is big, thereby breaking with the true dependence. The marginal distributions are not violated, as the points are in the 10th and 90th percentile, respectively, which is not too extreme. With feature 4, on the other hand, the situation is reversed, and it is anomalous because it does not fit with the true marginal distribution. The dependence is not violated since it is only related to feature 3. This dependence is such that it is likely that features 3 and 4 simultaneously have larger realisations, which is the case with the indicated point. It can be seen in Figure 4(b) that the indicated point is out-of-distribution simply because its 4th feature is too large.

## B.2 DIAGNOSTIC PLOTS

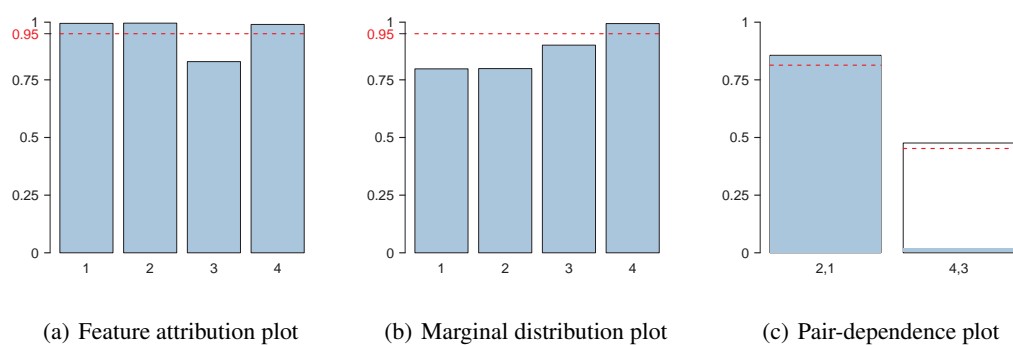

(a) Feature attribution plot  (b) Marginal distribution plot  (c) Pair-dependence plot

Figure 5: Diagnostic plots for explainability and feature attribution, applied to the situation of Figure 4, assess the indicated point in relation to the depicted distribution: 5(a) shows how much each feature of the indicated point violates the learned distribution of the training sample. 5(b) illustrates violations of the respective learned marginal distributions. Lastly, 5(c) depicts violations of the individual pair-copulas. The $95\%$ quantile is indicated in each plot by the dashed red line.

For a more intuitive understanding of why an observation is labeled as an outlier, we provide several diagnostic plots, depicted in Figure 5, by reference to the situation of Figure 4. The 'Feature attribution plot' 5(a) shows the quantiles of the feature scores relative to the training data. Observe that features 1,2 and 4 exceed the threshold of the $95\%$-quantile, indicated by the red dashed line. The 'Marginal distribution plot' 5(b) shows the quantiles of the marginal scores $m_i$ relative to the training data. As only the 4th feature exceeds the threshold, it can be inferred that the first two features are anomalous because they do not fit with the learned true structure. The 4th feature is anomalous because it does not fit with its marginal distribution.

Additionally, the 'Pair-dependence plot' shows how much each individual aspect of the dependence, as captured by each pair-copula, was violated. The maximal height of each bar is scaled by the parameter $\max_{ij}$, which quantifies the suitability of the corresponding copula $C_{i,j}$ to identify a new observation as an outlier, measured on a scale from 0 to 1. Only two bars are shown in Figure 5(c),

they correspond to the dependence between the 1st and 2nd feature and between the 3rd and 4th feature. VC-BOD correctly identifies those as the only relevant dependencies. It does so even though the truncation level was set to $T = 3$. Furthermore, it is evident that the point deviates from the established dependency structure between the 1st and 2nd features while aligning with the dependence observed between the 3rd and 4th feature. Additionally, the maximum heights of the bars are logical, as the dependence between the 1st and 2nd feature is more pronounced, making it more effective for outlier detection. This plot is particularly useful in higher dimensions to quickly identify pairs of features that behave in unusual ways, regardless of their marginal distributions.

### B.3 EMPIRICAL EVALUATION OF THE EXPLAINABILITY FRAMEWORK

In this subsection, we shift focus from theoretical examples to assessing the empirical capabilities of the explainability framework by applying it to a real-world dataset. Specifically, we use the wine dataset, which is also included in the benchmark dataset collection which was used for the empirical study in Section 6. The wine dataset comprises $n = 129$ observations, of which 119 are normal and 10 are labeled as outliers. It contains $d = 13$ variables, which are summarized in Table 1.

Table 1: Wine dataset variable descriptions.

| #Variable | Variable Name | Short Description |
|---|---|---|
| 1 | Alcohol | Alcohol content of the wine |
| 2 | Malicacid | Malic acid concentration |
| 3 | Ash | Ash content in the wine |
| 4 | Alcalinity_of_ash | Alkalinity of ash in the wine |
| 5 | Magnesium | Magnesium content |
| 6 | Total_phenols | Total phenol content |
| 7 | Flavanoids | Flavonoid phenol content |
| 8 | Nonflavanoid_phenols | Non-flavonoid phenol content |
| 9 | Proanthocyanins | Proanthocyanin content |
| 10 | Color_intensity | Intensity of the wine's color |
| 11 | Hue | Hue of the wine |
| 12 | OD280_OD315_of_diluted_wines | Optical density ratio at 280/315 nm |
| 13 | Proline | Proline amino acid concentration |

For this experiment, we train VC-BOD with a truncation level of $T = 3$ (VC-BOD T3) using the full set of normal observations as training data and then evaluate its performance on the outliers. The results show that 7 out of 10 outliers exceed the quantile threshold of $\tau = 0.95$, correctly identifying them as anomalies. The remaining three outliers are ranked in the 85th, 91st, and 93rd percentiles relative to the scores of the training data. Although these points fall below the suggested threshold of $\tau = 0.95$, their high percentile rankings further demonstrate the effectiveness of VC-BOD.

Figure 6 presents the diagnostic plots applied to one of the outliers correctly identified using $\tau = 0.95$. Refer to Subsection B.2 for a detailed explanation of the diagnostic plots' functionality. The feature attribution plot 6(a) reveals that features 1, 10, and 13 are the most influential in classifying this specific point. Combining this with insights from the marginal distribution plot 6(b) and the pair-dependence plot 6(c), we observe that features 13 and 1 primarily deviate from their marginal distributions, while feature 10 exhibits significant violations in its dependence relationships with other features. This is particularly evident from bars 1, 4, and 6 in Figure 6(c).

This example demonstrates the effectiveness of VC-BOD in providing a fast and intuitive understanding of why points are classified as outliers, even in higher-dimensional feature spaces. Additionally, the results underscore the potential importance of incorporating dependencies captured in higher-order trees of the vine copula, as evidenced by the presence of bar 6 in Figure 6(c).

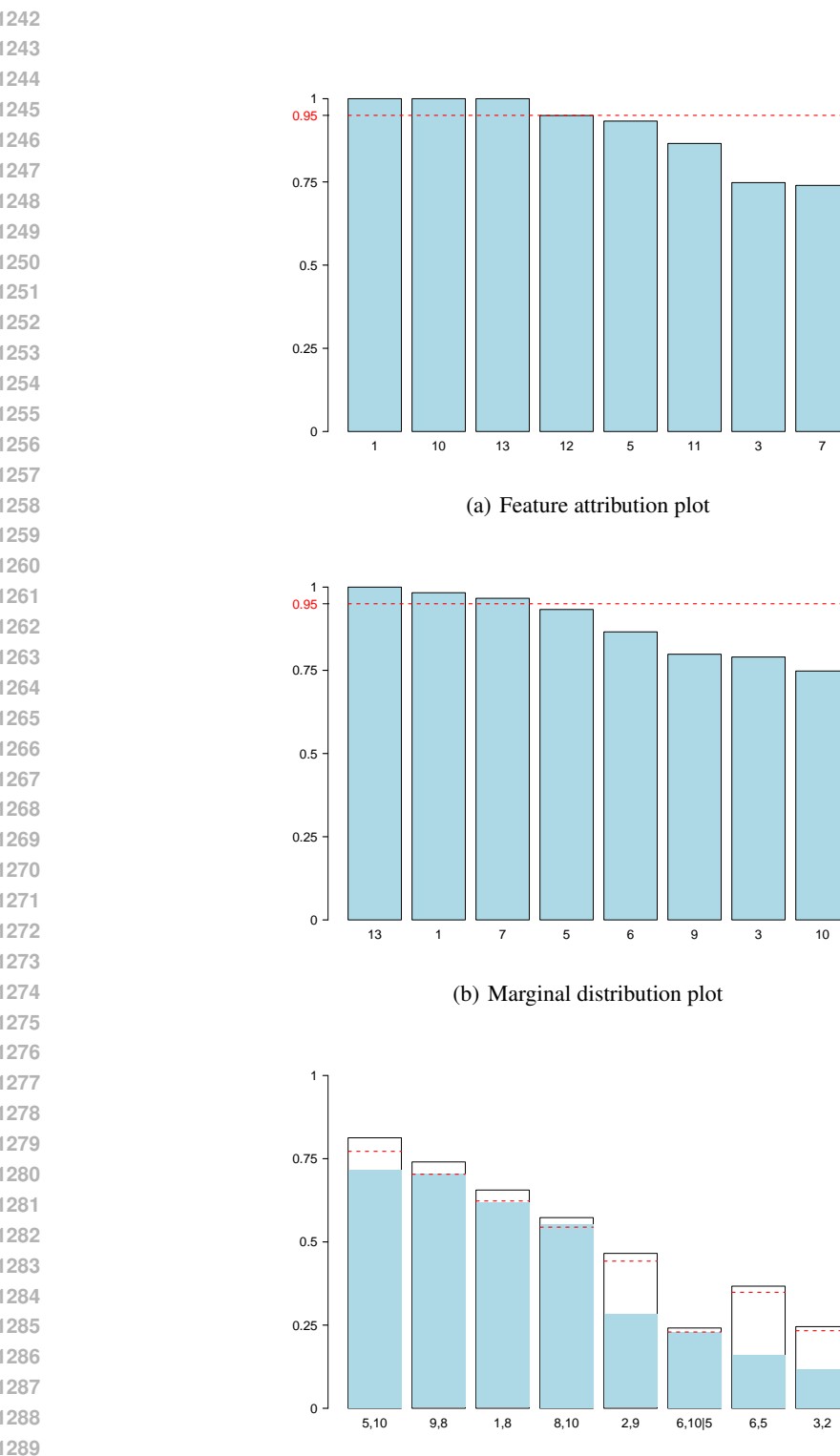

(a) Feature attribution plot

(b) Marginal distribution plot

(c) Pair-dependence plot

Figure 6: Diagnostic plots, see Figure 5 for details, applied to a true outlier of the 'wine' dataset. Different to Figure 5, each plots shows in descending order the 8 most influential features (or interactions) to keep the plots clear.

# C DATASETS

## C.1 SYNTHETIC DATASET

The dataset used in Section 6.1 is constructed as an R-vine, see, e.g., Czado (2019). For this, one needs to specify the tree structure and pair-copulas for each tree, as well as all marginal distributions. In the present case, the first tree connects features 1 and 2 and features 2 and 3. In both cases, the pair-copula is chosen as a Gaussian copula with parameter $\Theta = 0.4$. The second tree connects features 1 and 3, given feature 2. The connecting pair-copula is a Clayton copula with parameter $\Theta = 1.3$, rotated by 90 degrees. All three marginal distributions are chosen as the standard normal distribution.

## C.2 EMPIRICAL DATASETS

In Table 2 we provide an overview of the datasets used in our empirical evaluation. Column $n$ lists the total number of observations, and column $d$ shows the number of features per observation. Column *Outliers* provides the number of observations labeled as outliers, and column %*Outliers* indicates their percentage relative to the total number of observations.

Table 2: Datasets overview. A '*' behind the dataset name indicates that only 20% of the training data was used during training due to hardware constraints.

| Dataset | $n$ | $d$ | Outliers | %Outliers |
|---|---|---|---|---|
| Wine | 129 | 13 | 10 | 7.7% |
| Lympho | 148 | 18 | 6 | 4.1% |
| Glass | 214 | 9 | 9 | 4.2% |
| Vertebral | 240 | 6 | 30 | 12.5% |
| WBC | 278 | 30 | 21 | 5.6% |
| Ecoli | 336 | 7 | 9 | 2.6% |
| Ionosphere | 351 | 33 | 126 | 36% |
| Arrhythmia | 452 | 274 | 66 | 15% |
| BreastW | 683 | 9 | 239 | 35% |
| Pima | 768 | 8 | 268 | 35% |
| Vowels | 1456 | 12 | 50 | 3.4% |
| Letter Recognition | 1600 | 32 | 100 | 6.25% |
| Cardio | 1831 | 21 | 176 | 9.6% |
| Seismic | 2584 | 24 | 170 | 6.5% |
| Musk | 3062 | 166 | 97 | 3.2% |
| Speech | 3686 | 400 | 61 | 1.65% |
| Thyroid | 3772 | 6 | 93 | 2.5% |
| Abalone | 4177 | 9 | 29 | 0.69% |
| Optdigits | 5216 | 64 | 150 | 3% |
| Satimage-2 | 5803 | 36 | 71 | 1.2% |
| Satellite | 6435 | 36 | 2036 | 32% |
| Pendigits | 6870 | 16 | 156 | 2.27% |
| Annthyroid | 7200 | 6 | 534 | 7.42% |
| Mnist | 7603 | 100 | 700 | 9.2% |
| Mammography | 11183 | 6 | 260 | 2.32% |
| Shuttle | 49097 | 9 | 3511 | 7% |
| Mulcross* | 262144 | 4 | 26214 | 10% |
| ForestCover | 286048 | 10 | 2747 | 0.9% |
| Campaign | 41188 | 62 | 4640 | 11.3% |
| Fraud* | 284807 | 29 | 492 | 0.17% |
| Backdoor | 95329 | 196 | 2329 | 2.44% |

# D    EMPIRICAL EVALUATION

In this section, we provide a detailed overview of all the results of all the experiments we conducted. We further provide detailed performance metrics for all considered OD methods, which we obtained from Thimonier et al. (2024). In Table 3, we display the F1 metrics for our model, in Table 4 for all classical models and in Table 5 for all MLP-based models. For each method, we provide the total average F1-score and the standard deviation.

Table 3: F1-score for three variants of VC-BOD at truncation levels 0, 1, and 3, referred to as T0, T1, and T3, respectively. We highlight the best metric for each dataset in bold.

| Dataset | VC-BOD T0 | VC-BOD T1 | VC-BOD T3 |
|---|---|---|---|
| Wine | 65.75±9.19 | **68.25**±12.02 | 64.50±12.24 |
| Lympho | 90.83±9.82 | **92.50**±8.29 | 89.17±10.90 |
| Glass | 16.67±6.09 | 15.00±5.30 | **17.22**±6.06 |
| Vertebral | 13.83±2.05 | 14.50±5.80 | **16.25**±4.67 |
| Wbc | **70.24**±3.32 | 66.31±3.26 | 67.98±3.37 |
| Ecoli | 45.28±10.36 | 67.78±11.33 | **68.33**±10.11 |
| Ionosphere | 79.70±2.49 | 88.19±1.05 | **88.55**±1.05 |
| Arrhythmia | **62.01**±1.71 | 61.67±1.36 | 61.59±1.54 |
| Breastw | 97.08±0.64 | **97.10**±0.63 | 97.08±0.62 |
| Pima | 66.64±1.21 | **68.95**±1.44 | 68.56±1.45 |
| Vowels | 23.45±2.19 | 52.35±3.13 | **56.47**±3.34 |
| Letter | 17.62±1.77 | 34.80±1.87 | **39.38**±2.41 |
| Cardio | **64.28**±3.16 | 59.77±2.20 | 60.06±2.63 |
| Seismic | **29.41**±1.46 | 27.00±2.05 | 25.96±1.70 |
| Musk | **100.00**±0.00 | **100.00**±0.00 | **100.00**±0.00 |
| Speech | **4.80**±0.43 | **4.80**±0.43 | **4.80**±0.43 |
| Thyroid | **78.04**±2.22 | 70.99±2.10 | 74.87±3.17 |
| Abalone | 56.67±0.00 | **68.58**±2.88 | 66.50±2.88 |
| Optdigit | 5.32±0.84 | **16.98**±3.76 | 16.29±3.68 |
| Satimage | 83.10±1.22 | 93.79±1.40 | **94.28**±1.40 |
| Satellite | **75.64**±0.56 | 75.27±0.55 | 74.11±0.54 |
| Pendigit | **41.41**±1.15 | 32.04±1.90 | 31.39±2.69 |
| Annthyroid | 49.28±1.02 | 47.94±0.94 | **50.99**±1.14 |
| Mnist | 17.99±0.80 | **29.27**±1.00 | 29.05±1.08 |
| Mammo | 43.63±1.70 | **57.06**±0.84 | 54.38±1.47 |
| Shuttle | 97.68±0.28 | 98.44±0.12 | **98.56**±0.15 |
| Mullcr | 99.98±0.00 | **100.00**±0.00 | **100.00**±0.00 |
| Forest | 23.75±0.42 | 39.10±0.59 | **46.85**±0.73 |
| Campaign | **49.55**±0.25 | 45.76±0.34 | 47.30±1.02 |
| Fraud | 46.44±0.51 | 57.52±1.22 | **58.54**±1.02 |
| Backdoor | 19.62±5.99 | 33.34±13.68 | **36.68**±15.00 |
| mean | 52.76 | 57.58 | 58.24 |
| mean std | 2.3 | 3.0 | 3.2 |

Table 4: F1-score (↑) for all classical models. We highlight the best metric for each dataset in bold.

| Method | COPOD | IForest | KNN | PIDForest | RRCF | VC-BOD T3 |
|---|---|---|---|---|---|---|
| Wine | 60.00±4.50 | 64.00±12.80 | **94.00**±4.90 | 50.00±6.40 | 69.00±11.40 | 64.50±12.24 |
| Lympho | 85.00±5.00 | 71.70±7.60 | 80.00±11.70 | 70.00±0.00 | 36.70±18.00 | **89.17**±10.90 |
| Glass | 11.10±0.00 | 11.10±0.00 | 11.10±9.70 | 8.90±6.00 | 15.60±13.30 | **17.22**±6.06 |
| Vertebral | 1.70±1.70 | 13.00±3.80 | 10.00±4.50 | 12.00±5.20 | 8.00±4.80 | **16.25**±4.67 |
| Wbc | **71.40**±0.00 | 70.00±3.70 | 63.80±2.30 | 65.70±3.70 | 54.80±6.10 | 67.98±3.37 |
| Ecoli | 25.60±11.20 | 58.90±22.20 | **77.80**±3.30 | 25.60±11.20 | 28.90±11.30 | 68.33±10.11 |
| Ionosphere | 70.80±1.80 | 80.80±2.10 | **88.60**±1.60 | 67.10±3.90 | 72.00±1.80 | 88.55±1.05 |
| Arrhythmia | 58.20±1.40 | 60.90±3.30 | **61.80**±2.20 | 22.70±2.50 | 50.60±3.30 | 61.59±1.54 |
| Breastw | 96.40±0.60 | **97.20**±0.50 | 96.00±0.70 | 70.60±7.60 | 63.00±1.80 | 97.08±0.62 |
| Pima | 62.30±1.10 | **69.60**±1.20 | 65.30±1.00 | 65.90±2.90 | 55.40±1.70 | 68.56±1.45 |
| Vowels | 4.80±1.00 | 25.80±4.70 | **64.40**±3.70 | 23.20±3.20 | 18.00±4.60 | 56.47±3.34 |
| Letter | 12.90±0.70 | 15.60±3.30 | **45.00**±2.60 | 14.20±2.30 | 17.40±2.20 | 39.38±2.41 |
| Cardio | 65.00±1.40 | **73.50**±4.10 | 67.60±0.90 | 43.00±2.50 | 43.90±2.70 | 60.06±2.63 |
| Seismic | 29.20±1.30 | **73.90**±1.50 | 30.60±1.40 | 29.20±1.60 | 24.10±3.20 | 25.96±1.70 |
| Musk | 49.60±1.20 | 52.00±15.30 | **100.00**±0.00 | 35.40±0.00 | 38.40±6.50 | **100.00**±0.00 |
| Speech | 3.30±0.00 | 4.90±1.90 | **5.10**±1.00 | 2.00±1.90 | 3.90±2.80 | 4.80±0.43 |
| Thyroid | 30.80±0.50 | **78.90**±2.70 | 57.30±1.30 | 72.00±3.20 | 31.90±4.70 | 74.87±3.17 |
| Abalone | 50.30±6.40 | 53.40±1.70 | 43.40±4.80 | 58.60±1.60 | 36.90±6.40 | **66.50**±2.88 |
| Optdigit | 3.00±0.30 | 15.80±4.30 | **90.00**±1.20 | 22.50±16.80 | 1.30±0.70 | 16.29±3.68 |
| Satimage | 77.90±0.90 | 86.50±1.70 | 93.80±1.20 | 35.50±0.40 | 47.90±3.40 | **94.28**±1.40 |
| Satellite | 56.70±0.20 | 69.60±0.50 | **76.30**±0.40 | 46.90±3.70 | 55.40±1.30 | 74.11±0.54 |
| Pendigit | 34.90±0.60 | 52.10±6.40 | **91.00**±1.40 | 44.60±5.30 | 16.30±2.60 | 31.39±2.69 |
| Annthyroid | 31.50±0.50 | 57.30±1.30 | 37.80±0.60 | **65.40**±2.70 | 32.10±0.80 | 50.99±1.14 |
| Mnist | 38.50±0.40 | 51.20±2.50 | **69.40**±0.90 | 32.60±5.70 | 33.50±1.70 | 29.05±1.08 |
| Mammo | 53.40±0.90 | 39.00±3.30 | 38.80±1.50 | 28.10±4.30 | 27.10±1.90 | **54.38**±1.47 |
| Shuttle | 96.00±0.00 | 96.40±0.80 | 97.30±0.20 | 70.70±1.00 | 32.00±2.20 | **98.56**±0.15 |
| Mullcr | 66.00±0.10 | 99.10±0.50 | **100.00**±0.00 | 67.40±2.10 | **100.00**±0.00 | **100.00**±0.00 |
| Forest | 18.20±0.20 | 11.10±1.60 | **92.10**±0.30 | 8.10±2.80 | 9.90±1.50 | 46.85±0.73 |
| Campaign | **49.50**±0.10 | 42.40±1.00 | 41.60±0.40 | 42.40±0.20 | 36.60±0.10 | 47.30±1.02 |
| Fraud | 44.70±0.90 | 30.30±3.70 | **60.50**±1.50 | 41.00±0.90 | 17.10±0.40 | 58.54±1.02 |
| Backdoor | 13.40±0.40 | 3.80±1.20 | **88.50**±0.10 | 3.40±0.20 | 24.50±0.10 | 36.68±15.00 |
| mean | 44.3 | 52.6 | 65.8 | 40.2 | 35.6 | 58.2 |
| mean std | 1.5 | 3.9 | 2.2 | 3.6 | 4.0 | 3.2 |

Table 5: F1-score (↑) for the MLP based models. We highlight the best metric for each dataset in bold.

| Method | GOAD | NeuTraL. | Inter.Cont. | NPT-AD | Transformer | VC-BOD T3 |
|---|---|---|---|---|---|---|
| Wine | 67.00±9.40 | 78.20±4.50 | **90.00**±6.30 | 72.50±7.70 | 23.50±7.90 | 64.50±12.24 |
| Lympho | 68.30±13.00 | 20.00±18.70 | 86.70±6.00 | **94.20**±7.90 | 88.30±7.60 | 89.17±10.90 |
| Glass | 12.70±3.90 | 9.00±4.40 | **27.20**±10.60 | 26.20±10.90 | 14.40±6.10 | 17.22±6.06 |
| Vertebral | 16.30±9.60 | 3.80±1.20 | **26.00**±7.70 | 20.30±4.80 | 12.30±5.20 | 16.25±4.67 |
| Wbc | 66.20±2.90 | 60.90±5.60 | 67.60±3.60 | 67.30±1.70 | 66.40±3.20 | **67.98**±3.37 |
| Ecoli | 61.40±31.70 | 7.00±7.10 | 70.00±7.80 | **77.70**±0.10 | 75.00±9.90 | 68.33±10.11 |
| Ionosphere | 83.40±2.60 | 90.60±2.40 | **93.20**±1.30 | 92.70±0.60 | 88.10±2.80 | 88.55±1.05 |
| Arrhythmia | 52.00±2.30 | 59.50±2.60 | **61.80**±1.80 | 60.40±1.40 | 59.80±2.20 | 61.59±1.54 |
| Breastw | 96.00±0.60 | 91.80±1.30 | 96.10±0.70 | 95.70±0.30 | 96.70±0.30 | **97.08**±0.62 |
| Pima | 66.00±3.10 | 60.30±1.40 | 59.10±2.20 | **68.80**±0.60 | 65.60±2.00 | 68.56±1.45 |
| Vowels | 31.10±4.20 | 10.00±6.20 | **90.80**±1.60 | 88.70±1.60 | 28.70±8.00 | 56.47±3.34 |
| Letter | 20.70±1.70 | 5.70±0.80 | 62.80±2.40 | **71.40**±1.90 | 41.50±6.20 | 39.38±2.41 |
| Cardio | **78.60**±2.50 | 45.50±4.30 | 71.00±2.40 | 78.10±0.10 | 68.80±2.80 | 60.06±2.63 |
| Seismic | 24.10±1.00 | 11.80±4.30 | 20.70±1.90 | **26.20**±0.70 | 19.10±5.70 | 25.96±1.70 |
| Musk | **100.00**±0.00 | 99.00±0.00 | **100.00**±0.00 | **100.00**±0.00 | **100.00**±0.00 | **100.00**±0.00 |
| Speech | 4.80±2.30 | 4.70±1.40 | 5.20±1.20 | **9.30**±0.80 | 6.80±1.90 | 4.80±0.43 |
| Thyroid | 72.50±2.80 | 69.40±1.40 | 76.80±1.20 | **77.00**±0.60 | 55.50±4.80 | 74.87±3.17 |
| Abalone | 57.60±2.20 | 53.20±4.00 | **68.70**±2.30 | 59.70±0.10 | 42.50±7.80 | 66.50±2.88 |
| Optdigit | 0.30±0.30 | 16.20±7.30 | **66.30**±10.10 | 62.00±2.70 | 61.10±4.70 | 16.29±3.68 |
| Satimage | 90.70±0.70 | 92.30±1.90 | 92.40±0.70 | **94.80**±0.80 | 89.00±4.10 | 94.28±1.40 |
| Satellite | 64.20±0.80 | 71.60±0.60 | 73.20±1.60 | **74.60**±0.70 | 65.60±3.30 | 74.11±0.54 |
| Pendigit | 40.10±5.00 | 69.80±8.70 | 82.30±4.50 | **92.50**±1.30 | 35.40±10.90 | 31.39±2.69 |
| Annthyroid | 50.30±6.30 | 44.10±2.30 | 45.40±1.80 | **57.70**±0.60 | 29.90±1.50 | 50.99±1.14 |
| Mnist | 66.90±1.30 | 84.80±0.50 | **85.90**±0.00 | 71.80±0.30 | 56.70±5.70 | 29.05±1.08 |
| Mammo | 33.70±6.10 | 19.20±2.40 | 29.40±1.40 | 43.60±0.50 | 17.40±2.20 | **54.38**±1.47 |
| Shuttle | 73.50±5.10 | 97.90±0.20 | 98.40±0.10 | 98.20±0.30 | 85.30±9.80 | **98.56**±0.15 |
| Mullcr | 99.70±0.80 | 96.30±10.50 | **100.00**±0.00 | **100.00**±0.00 | **100.00**±0.00 | **100.00**±0.00 |
| Forest | 0.10±0.20 | 51.60±8.20 | 44.00±4.10 | **58.00**±10.00 | 21.30±3.10 | 46.85±0.73 |
| Campaign | 16.20±1.80 | 42.10±1.70 | 46.80±1.40 | **49.80**±0.30 | 47.00±1.90 | 47.30±1.02 |
| Fraud | 53.10±10.20 | 24.30±7.80 | 57.90±2.80 | 58.10±3.20 | 54.30±5.20 | **58.54**±1.02 |
| Backdoor | 12.70±2.90 | 84.40±1.80 | **86.60**±0.10 | 84.10±0.10 | 85.80±0.60 | 36.68±15.00 |
| mean | 51.0 | 50.8 | 67.2 | 68.8 | 54.9 | 58.2 |
| mean std | 4.4 | 4.0 | 2.9 | 2.0 | 4.3 | 3.2 |

Table 6: Outlier detection AUC-Scores ($\uparrow$) for T0,T1,T3.

| Method | VC-BOD T0 | VC-BOD T1 | VC-BOD T3 |
|---|---|---|---|
| Wine | 80.02±5.36 | **81.48**±7.01 | 79.29±7.14 |
| Lympho | 95.03±5.33 | **95.93**±4.50 | 94.13±5.91 |
| Glass | 54.69±3.31 | 53.79±2.88 | **54.99**±3.29 |
| Vertebral | 44.61±1.32 | 45.04±3.73 | **46.16**±3.00 |
| Wbc | **83.37**±1.86 | 81.18±1.82 | 82.11±1.88 |
| Ecoli | 71.14±5.46 | 83.00±5.98 | **83.30**±5.33 |
| Ionosphere | 78.53±2.63 | 87.52±1.11 | **87.89**±1.11 |
| Arrhythmia | **74.51**±1.15 | 74.28±0.91 | 74.23±1.03 |
| Breastw | 96.97±0.66 | **96.99**±0.65 | 96.96±0.65 |
| Pima | 65.44±1.25 | **67.83**±1.49 | 67.43±1.50 |
| Vowels | 59.00±1.17 | 74.48±1.67 | **76.70**±1.80 |
| Letter | 53.32±1.00 | 63.05±1.06 | **65.65**±1.36 |
| Cardio | **78.34**±1.91 | 75.61±1.33 | 75.78±1.59 |
| Seismic | **59.73**±0.84 | 58.36±1.17 | 57.76±0.97 |
| Musk | **100.00**±0.00 | **100.00**±0.00 | **100.00**±0.00 |
| Speech | **50.80**±0.22 | **50.80**±0.22 | **50.80**±0.22 |
| Thyroid | **88.46**±1.16 | 84.76±1.10 | 86.80±1.66 |
| Abalone | 77.65±0.00 | **83.79**±1.48 | 82.72±1.49 |
| Optdigit | 49.85±0.44 | **56.03**±1.99 | 55.67±1.95 |
| Satimage | 91.34±0.62 | 96.82±0.72 | **97.09**±0.73 |
| Satellite | **76.55**±0.54 | 76.19±0.53 | 75.07±0.52 |
| Pendigit | **69.34**±0.60 | 64.44±1.00 | 64.10±1.41 |
| Annthyroid | 70.58±0.59 | 69.80±0.54 | **71.57**±0.66 |
| Mnist | 50.68±0.48 | **57.47**±0.60 | 57.33±0.65 |
| Mammo | 70.47±0.89 | **77.51**±0.44 | 76.11±0.77 |
| Shuttle | 98.66±0.16 | 99.10±0.07 | **99.17**±0.08 |
| Mullcr | 99.99±0.00 | **100.00**±0.00 | **100.00**±0.00 |
| Forest | 61.14±0.21 | 68.96±0.30 | **72.91**±0.37 |
| Campaign | **68.37**±0.15 | 66.00±0.21 | 66.96±0.64 |
| Fraud | 73.13±0.25 | 78.69±0.61 | **79.20**±0.51 |
| Backdoor | 57.80±3.15 | 65.00±7.18 | **66.76**±7.88 |
| mean | 72.56 | 75.28 | 75.63 |
| mean std | 1.3 | 1.7 | 1.8 |

### D.1 STATISTICAL EVALUATION OF PERFORMANCE

To evaluate if there is a significant difference in the performance of VC-BOD, as indicated by average rank across the datasets, against benchmark models, we conduct a statistical analysis using the Friedman test followed by the Nemenyi post-hoc test, as described in Liu et al. (2019); Campos et al. (2016). The null hypothesis of the Friedman test assumes no significant difference between algorithms; it is rejected if the p-value is below the chosen significance level, indicating at least two algorithms differ significantly. The Nemenyi test is used to identify specific pairs with significant differences.

Our analysis yields several noteworthy observations. The Friedman-Nemenyi test does not confirm that our method significantly outperforms the KNN algorithm. However, it also reveal that none of the current state-of-the-art deep learning methods demonstrate statistically significant superiority over our approach in terms of rank performance. This finding suggests that our method achieves competitive performance compared to existing state-of-the-art techniques in outlier detection, including those based on deep learning.

Table 7 presents the pairwise comparison results among the models based on the Nemenyi test. In the table, an upward arrow ($\uparrow$) indicates that the model in the row significantly outperforms the model in the column at a confidence level of $0.1$, while a downward arrow ($\downarrow$) indicates the opposite. Double arrows ($\uparrow\uparrow$ or $\downarrow\downarrow$) denote statistical significance at a confidence level $0.05$.

Table 7: Comparison of Models Based on Nemenyi Test

| | COPOD | IForest | KNN | PIDForest | RRCF | GOAD | NeuTraL | Inter.Cont. | NPT-AD | Transformer | VC-BOD T0 | VC-BOD T1 | VC-BOD T3 |
|---|---|---|---|---|---|---|---|---|---|---|---|---|---|
| COPOD | - | | $\downarrow\downarrow$ | | | | | $\downarrow\downarrow$ | $\downarrow\downarrow$ | | $\downarrow$ | $\downarrow\downarrow$ | $\downarrow\downarrow$ |
| IForest | | - | | $\uparrow$ | $\uparrow\uparrow$ | | | $\downarrow\downarrow$ | $\downarrow\downarrow$ | | | | |
| KNN | $\uparrow\uparrow$ | | - | $\uparrow\uparrow$ | $\uparrow\uparrow$ | | $\uparrow\uparrow$ | | | | | | |
| PIDForest | | $\downarrow$ | $\downarrow\downarrow$ | - | | | | $\downarrow\downarrow$ | $\downarrow\downarrow$ | | $\downarrow\downarrow$ | $\downarrow\downarrow$ | $\downarrow\downarrow$ |
| RRCF | $\downarrow\downarrow$ | $\downarrow\downarrow$ | $\downarrow\downarrow$ | | - | | | $\downarrow\downarrow$ | $\downarrow\downarrow$ | $\downarrow\downarrow$ | $\downarrow\downarrow$ | $\downarrow\downarrow$ | $\downarrow\downarrow$ |
| GOAD | | | | | | - | | $\downarrow\downarrow$ | $\downarrow\downarrow$ | | | | |
| NeuTraL | | | $\downarrow\downarrow$ | | | | - | $\downarrow\downarrow$ | $\downarrow\downarrow$ | | | $\downarrow\downarrow$ | $\downarrow\downarrow$ |
| Inter.Cont. | $\uparrow\uparrow$ | | | $\uparrow\uparrow$ | $\uparrow\uparrow$ | $\uparrow\uparrow$ | $\uparrow\uparrow$ | - | | $\uparrow$ | | | |
| NPT-AD | $\uparrow\uparrow$ | $\uparrow\uparrow$ | | $\uparrow\uparrow$ | $\uparrow\uparrow$ | $\uparrow\uparrow$ | $\uparrow\uparrow$ | | - | $\uparrow\uparrow$ | | | |
| Transformer | | | | $\uparrow\uparrow$ | | | | $\downarrow$ | $\downarrow\downarrow$ | - | | | |
| VC-BOD T0 | $\uparrow$ | | | $\uparrow\uparrow$ | $\uparrow\uparrow$ | | | | | | - | | |
| VC-BOD T1 | $\uparrow\uparrow$ | | | $\uparrow\uparrow$ | $\uparrow\uparrow$ | | $\uparrow\uparrow$ | | | | | - | |
| VC-BOD T3 | $\uparrow\uparrow$ | | | $\uparrow\uparrow$ | $\uparrow\uparrow$ | | $\uparrow\uparrow$ | | | | | | - |

## D.2 COMPUTATIONAL TIME

Table 8: Runtime in seconds for the training and inference time for each dataset. The training and inference runtimes correspond to the average training and inference times of the model over 40 runs. Due to hardware constraints, the runtimes for 'Fraud' and 'Mullcorss' were determined using only 20% of the total training data and 50% for inference. The results were achieved using 20 CPU cores.

| Dataset | VCBOD T0 train | VCBOD T0 inference | VCBOD T1 train | VCBOD T1 inference | VCBOD T3 train | VCBOD T3 inference |
|---|---|---|---|---|---|---|
| Wine | 1.55 | 1.60 | 3.81 | 3.20 | 4.02 | 3.19 |
| Lympho | 1.54 | 1.57 | 3.31 | 2.46 | 3.30 | 2.33 |
| Glass | 1.93 | 1.95 | 4.49 | 3.89 | 4.57 | 3.85 |
| Vertebral | 0.36 | 0.41 | 1.09 | 0.77 | 1.03 | 0.65 |
| Wbc | 1.71 | 1.72 | 4.31 | 3.37 | 5.07 | 3.39 |
| Ecoli | 1.83 | 1.87 | 4.03 | 3.75 | 4.20 | 3.72 |
| Ionosph. | 1.50 | 1.50 | 4.13 | 3.05 | 5.23 | 3.22 |
| Arrhyth. | 1.67 | 1.48 | 7.33 | 3.33 | 11.41 | 3.70 |
| Breastw | 0.53 | 0.65 | 1.07 | 0.62 | 1.05 | 0.61 |
| Pima | 1.42 | 1.52 | 3.22 | 3.01 | 3.26 | 3.01 |
| Vowels | 1.57 | 1.71 | 3.94 | 3.33 | 4.54 | 3.30 |
| Letter | 1.86 | 2.10 | 4.66 | 4.06 | 5.63 | 4.19 |
| Cardio | 1.02 | 1.22 | 2.57 | 2.38 | 3.21 | 2.37 |
| Seismic | 1.53 | 1.77 | 3.31 | 3.21 | 3.50 | 3.21 |
| Musk | 3.58 | 5.10 | 12.19 | 10.48 | 21.84 | 14.43 |
| Speech | 5.13 | 9.75 | 18.48 | 10.10 | 39.28 | 10.20 |
| Thyroid | 1.66 | 1.86 | 3.99 | 3.67 | 4.79 | 3.65 |
| Abalone | 1.40 | 1.53 | 3.42 | 3.10 | 3.71 | 3.29 |
| Optdigits | 1.88 | 2.51 | 13.27 | 4.59 | 89.91 | 4.81 |
| Satimage2 | 1.01 | 1.44 | 4.73 | 2.77 | 10.05 | 2.85 |
| Satellite | 2.15 | 2.89 | 27.39 | 5.53 | 67.71 | 5.64 |
| Pendigits | 1.03 | 1.29 | 10.84 | 2.65 | 44.65 | 2.85 |
| Annthyr. | 1.93 | 2.31 | 5.30 | 4.49 | 9.91 | 4.45 |
| Mnist | 1.42 | 2.73 | 3.61 | 3.44 | 7.61 | 3.57 |
| Mammo. | 1.90 | 3.34 | 16.97 | 5.41 | 23.79 | 5.44 |
| Shuttle | 0.84 | 3.05 | 71.08 | 4.46 | 187.53 | 4.95 |
| Mullcross | 1.03 | 3.97 | 12242.49 | 7.44 | 19754.35 | 8.9 |
| Forest | 4.16 | 9.96 | 2244.81 | 20.04 | 7012.54 | 22.09 |
| Campaign | 2.84 | 4.76 | 6.28 | 6.82 | 11.22 | 7.80 |
| Fraud | 1.76 | 15.80 | 127.04 | 24.75 | 337.30 | 30.23 |
| Backdoor | 13.63 | 21.32 | 26.01 | 28.72 | 53.51 | 37.40 |

