# OpenReview forum: "Decoupling Dependency Structures: Sklar’s theorem for explainable outlier detection"
_ICLR.cc/2025/Conference — Submitted to ICLR 2025_

### Official Review · Reviewer_adkm · 2024-10-31

**Soundness:** 3
**Presentation:** 3
**Contribution:** 3
**Rating:** 5
**Confidence:** 4

**Summary:**

The paper introduces a novel approach to outlier detection (OD) named Vine Copula-Based Outlier Detection (VC-BOD). VC-BOD leverages Sklar's theorem, vine copulas, and univariate kernel density estimators to separate marginal distributions from dependency structures for the purpose of detecting outliers. The model provides a closed-form equation for calculating an outlier score, enabling detailed explainability and feature attribution. It distinguishes between features that deviate from their own distributions and those affected by interactions with other features. The empirical evaluation shows that VC-BOD outperforms classical models and is competitive with deep learning models in terms of performance.

**Strengths:**

1. The paper builds on a strong theoretical foundation with Sklar's theorem, providing a solid basis for the proposed method.
2. VC-BOD offers a high level of explainability, which is a significant advantage over black-box deep learning models, especially in fields requiring transparent decision-making.
3. The paper demonstrates that VC-BOD achieves state-of-the-art performance, outperforming all classical models and competing well against deep learning models. The empirical evaluation is thorough, with a comprehensive analysis across 31 tabular datasets, benchmarking VC-BOD against existing methodologies.

**Weaknesses:**

1. While the paper claims VC-BOD can handle high-dimensional data, the computational complexity and training times could be prohibitive for very large datasets.
2. The method assumes that the data can be modeled using kernel density estimators and copulas, which may not always hold, especially for data with complex or non-smooth distributions.
3. The paper acknowledges that the theory described is not directly applicable to non-continuous features, which may limit the method's applicability in certain domains.
4. The paper does not explicitly address the potential for overfitting, especially when using the full dependency structure captured by the vine copula.

**Questions:**

See weaknesses.

---

> ### Author Response · Authors · 2024-11-17
>
> First, we would like to thank the reviewer for the positive feedback, for taking the time to read the paper thoroughly, and for identifying potential weaknesses. Below, we provide responses to questions and suggestions.
>
> ## Weaknesses
>
> 1. *While the paper claims VC-BOD can handle high-dimensional data, the computational complexity and training times could be prohibitive for very large datasets.
> We highly appriacte mentioning this point. As this issue was also raised by a different reviewer, we repeat the answer we gave there:*
>
> While it is true that this issue exists, we believe it is important to emphasize that it is not a theoretical limitation but rather a practical challenge stemming from the current implementation of copula methods, which are designed to operate solely on CPUs. Large datasets could likely benefit significantly from an implementation that leverages GPUs.
>
>
> 2. *The method assumes that the data can be modeled using kernel density estimators and copulas, which may not always hold, especially for data with complex or non-smooth distributions.*
>
> 3. *The paper acknowledges that the theory described is not directly applicable to non-continuous features, which may limit the method's applicability in certain domains.*
>
> We would like to address weaknesses 2 and 3 together.
>
> It is true that our method experiences some degree of information loss when non-continuous features are present in the dataset. This occurs because, in such cases, the feature outlier score for non-continuous features does not account for interactions with other features. However, we want to note that VC-BOD, in its current implementation, is capable of handling any number of non-continuous features. This ability, which is demonstrated in the empirical study in Section 6.2,  is not commonly found in density-based outlier detection methods.
>
>
> 4. *The paper does not explicitly address the potential for overfitting, especially when using the full dependency structure captured by the vine copula.*
>
> First, we would like to note that our approach involves using a truncated vine copula (truncated after tree 1 or 3 in the empirical study), meaning that we rarely model the entire distribution. More broadly, the issue of overfitting relates to how well the training data represents the true underlying distribution.
> In the context of VC-BOD, overfitting occurs when a specific region of the density—whether pertaining to a marginal distribution or a (conditional) bivariate aspect of the overall distribution—is assigned substantially more probability mass than it should receive based on the true underlying distribution. If this region is, in fact, a low-density area, it can impede VC-BOD’s ability to detect outliers that appear in this region.
>
> Would the reviewer find it beneficial to include this discussion on overfitting in the paper?
>
>
>
>
> We hope this response adequately addresses the reviewer’s concerns. We invite the reviewer to engage in further discussion regarding our paper.

---

### Official Review · Reviewer_5yhb · 2024-11-03

**Soundness:** 3
**Presentation:** 2
**Contribution:** 3
**Rating:** 5
**Confidence:** 4

**Summary:**

This paper proposes a vine-copula-based method for outlier detection. The proposed method learns how features are related in terms of a vine copula, which then enables us to see which features contribute most to the proposed outlier score. The experimental results show that the proposed method is highly competitive in practice.

**Strengths:**

- The high-level strategy proposed is neat and makes a lot of sense.
- The experimental results look very promising.
- The interpretability aspect is very straightforward to understand for this method, largely owing to existing vine copula theory.

**Weaknesses:**

- Right now, the text has a *lot* of English grammar issues/typos. Please proofread and/or use a grammar checker. Partly because of these English issues, I had trouble following Section 4, where the outlier score is defined. I would suggest cleaning up the exposition to make this section much clearer, especially as it is for your proposed method.
- I think the experimental results would benefit from trying to drill down on what specific kinds of datasets the proposed method works extremely well for, versus which ones it doesn't work as well on. While Figure 3 is indeed very informative in terms of performance at the aggregate level across datasets, I don't think it helps us explain when and why we should expect VC-BOD to work well. For example, a breakdown of how the different methods (the proposed ones truncated at different levels vs baselines) perform for datasets with low vs medium vs high number of dimensions could possibly be helpful.
- Related to my second point above, I think adding some sort of experiments that looks at how training dataset size and dataset dimensionality impact results could be helpful. My suspicion is that for the Gaussian kernel density estimator to work well, it already needs a decent number of samples, and then to estimate the dependency structure takes even more data (the first paragraph in the discussion section focuses on computation time, but here I'm talking about accuracy/how well we can even estimate the dependency structure). As an example, you could take one of the datasets that you are already using and that is sufficiently large, and intentionally train only using 5%, 10%, 15%, ..., 100% of the data, and see how performance changes as a function of how much training data is used. Obviously there's also an issue here where for a dataset that has features that are independent, the problem should actually be easier since there's no dependency to learn basically.
- The paper already mentions how the Gaussian kernel density estimator's bandwidth is chosen but the existing rules of thumb for choosing the bandwidth were not designed with outlier detection or vine copula estimation in mind (as far as I know; they were just focused on minimizing mean integrated square error in density estimation), and I'm wondering to what extent tuning the bandwidth affects results. Basically if I understand the setup here correctly, part of the issue is that the bandwidth choice has a downstream impact on learning the dependency structure (so that it could be that, for example, intentionally either over-smoothing or under-smoothing has a less negative effect on how well the dependency structure is learned in the vine copula---I don't have good intuition of whether you'd want over-smoothing or under-smoothing in this case). Adding an experiment to study this would be helpful.

Minor (I might not have caught all the typos but here are some):
- On line ~142 (definition of a copula), "uniform marginals on $[0; 1]$" should instead say "uniform marginals on $[0,1]$"
- On line 219 (second line of Section 4), "In this paragraph" should instead say "In this section"
- On lines 234, 235, and 243: the standard notation for an interval is usually of the form $[a,b]$ and not $[a;b]$ (for example, even when copulas are first defined, the notation used is correct on line ~142 and says "$C:[0,1]^d\rightarrow[0,1]$" --- notice that the domain and range use interval notation correctly)

**Questions:**

Please address the weakness points that I raised.

---

> ### Author Response · Authors · 2024-11-18
>
> Dear reviewer, first of all, thank you for reading our manuscript carefully and raising potential shortcomings and minor typing issues. A reply to the concerns can be found below.
>
> ## Weaknesses
>
> 1. *Right now, the text has a lot of English grammar issues/typos. Please proofread and/or use a grammar checker. Partly because of these English issues, I had trouble following Section 4, where the outlier score is defined. I would suggest cleaning up the exposition to make this section much clearer, especially as it is for your proposed method.*
>
> We appreciate the reviewer’s feedback regarding grammar and typo issues. However, after multiple rounds of proofreading, we were unable to identify the specific grammatical or linguistic challenges encountered during the review. We kindly request the reviewer to point out specific examples of these issues to help us address them more effectively. Additionally, we have included Section A.1 at the beginning of the Appendix providing a statistical derivation of our outlier score for further clarity.
>
>
> 2. *I think the experimental results would benefit from trying to drill down on what specific kinds of datasets the proposed method works extremely well for, versus which ones it doesn't work as well on. While Figure 3 is indeed very informative in terms of performance at the aggregate level across datasets, I don't think it helps us explain when and why we should expect VC-BOD to work well. For example, a breakdown of how the different methods (the proposed ones truncated at different levels vs baselines) perform for datasets with low vs medium vs high number of dimensions could possibly be helpful.*
>
> Defining a theoretically sound a priori criterion to evaluate the performance of VC-BOD or any other distribution-based outlier detection method solely based on the available training data of normal observations is challenging. This is primarily because the performance heavily depends on the distribution of outliers, which cannot be inferred during training under our experimental setup.
>
> However, it is reasonable to expect VC-BOD to perform better in low-dimensional settings. When working with truncated vine copulas in such settings, a greater portion of the overall distribution is captured, increasing the chance of correctly identifying true outliers. For a detailed explanation of this tradeoff, we kindly refer the reviewer to the newly added Section A.1 in the Appendix.
>
> We would also like to emphasize that all outlier detection methods are known to suffer from the curse of dimensionality, as established in [Zimek et al., 2012](https://doi.org/10.1002/sam.11161).
>
> Regarding weaknesses 3 and 4, we are currently conducting additional experiments to address the points raised. We will update our response with the results as soon as the experiments are completed.
>
> 5. Minor formatting issues
>
> Finally, we thank the reviewer for pointing out the formatting issues, which we hope to have successfully addressed.
>
> We invite the reviewer to engage in further discussion regarding our rebuttal.
>
> [Zimek et al., 2012](https://onlinelibrary.wiley.com/doi/abs/10.1002/sam.11161) Zimek, Arthur, Schubert, Erich, and Kriegel, Hans-Peter. "A survey on unsupervised outlier detection in high-dimensional numerical data." *Statistical Analysis and Data Mining: The ASA Data Science Journal*, 5(5), 363–387, 2012.

---

> ### Author Response · Authors · 2024-11-21
> **Response to Weakness 3 & 4**
>
> 4. *Related to my second point above, I think adding some sort of experiments that looks at how training dataset size and dataset dimensionality impact results could be helpful. My suspicion is that for the Gaussian kernel density...*
>
>
> We conducted an experiment following the design suggested by the reviewer to evaluate the impact of training set size on performance. While we occasionally observed a slight performance decline with reduced training set sizes, in many cases, no clear trend was evident.
>
> For example, considering the vertebral and campaign datasets, we analyzed performance when using 10% and 50% of the normal training data while keeping the test size fixed at 50%. For vertebral, there was a modest performance improvement, with accuracy increasing from 12.6% to 17.3% as the training size grew from 10% to 50%. In contrast, for campaign, performance remained stable around 64.4%, showing no notable effect of the training size on the outcome. Importantly, across all analyses, we found no significant differences attributable to the variation in training sizes for the chosen methods.
>
> Thus VC-BOD demonstrates strong and stable performance with minimal data requirements, which is a significant advantage. This robustness arises from our approach of analyzing numerous one- and two-dimensional aspects of the overall distribution. For a more detailed discussion, we refer to the newly added Section A.1 in the Appendix. While the vine copula structure may vary across different random seeds for the same dataset, VC-BOD remains highly robust due to its reliance on Kendall’s tau, a rank-based measure, for determining the tree structure. As a result, the structure of the first tree exhibits minimal variation. Notably, the structures in higher-order trees (e.g., Trees 2 and 3) exhibit more variability, as they involve conditional distributions influenced by copulas from the lower-order trees. However, the impact of these copulas on the overall outlier score is discounted, effectively mitigating the effect of this variation.
>
>
>
> 5. *The paper already mentions how the Gaussian kernel density estimator's bandwidth is chosen but the existing rules of thumb for choosing the bandwidth were not designed with outlier...*
>
> We acknowledge that the degree of smoothing applied in univariate kernel estimator can influence downstream tasks such as copula estimation and subsequent outlier detection. However, we argue that this effect is negligible. To support this claim, we conducted experiments on a subset of the datasets (due to computational constraints) using a semi-parametric approach to estimate the vine copula. Specifically, we employed the empirical distribution function (i.e., without smoothing) to transform the data points to the copula scale. The results revealed no significant differences compared to those reported in the paper, reinforcing our assertion.
>
>
> Having thoroughly addressed all the concerns raised by the reviewer, we look forward to receiving the reviewer's feedback.

---

> > ### Comment · Reviewer_5yhb · 2024-11-25
> >
> > **Overall:** Thanks for addressing some of my concerns. After reading all the reviews and the author responses to all reviews, I'm leaving my score the same. Overall, I still think that the paper has lingering issues and lacks polish.
> >
> > I think within the main paper, more clearly doing a breakdown of the performance across different kinds of datasets by dimensionality would really be helpful (even better would be to also do a breakdown that accounts or feature dependence). I understand that other outlier/anomaly detection methods can also run into curse of dimensionality issues, but I think it would be helpful to see how different methods cope with it (where it is possible that some do worse than others, for instance). Note that this does not even involve running new experiments. It just involves, for example, making a plot like Figure 3 where you restrict datasets to ones that have high dimensions vs ones that have low dimensions (or even breaking the datasets up into 3 groups of high, medium, and low dimensionality). Here, to be a little more explicit, within the statistics community, it is common to look at the ratio $d/n$ (or alternatively $n/d$) to gauge how high-dimensional a dataset is; you could threshold this ratio depending on the datasets that you are actually using to decide between high vs low dimensionality.
> >
> > **Some other issues I noticed after looking over the paper again:**
> > - In Section 2, I'd suggest just saying classical vs deep-learning-based instead of classical vs MLP, especially since a number of the methods mentioned use fancier neural net architectures than just an MLP.
> > - On page 6, regarding noncontinuous features: perhaps it would be helpful just being a bit clearer here about the specific kinds of noncontinuous features that actually show up and how they are handled. For example, if we encounter a binary feature, or we use one-hot encoding so that we effectively have binary features, then I don't see why a kernel density estimator makes sense for these features. Meanwhile, if the noncontinuous feature is a discrete ordinal/"leveled" feature, then sure I can see why a continuous approximation makes sense.
> >
> > **Regarding English issues/other typos, here are some (non-exhaustive, some minor and some that should more urgently be addressed):**
> > - Page 2, line ~90: The phrase "For the contexts of our work" is strangely worded ("contexts" should not be plural here) and actually this phrase could just be deleted (and the text would still make sense).
> > - Page 2, line ~93-94: "VC-BOD is to be classified [...]" can just say "VC-BOD is classified [...]"
> > - Page 4, line ~185-186: You don't need the word "so-called" before "trees" especially since trees are a well-known standard data structure in computer science; for example, one way to reword the sentence is: "This subclass of multivariate copula functions is constructed out of several bivariate copulas, which are arranged in successive levels; each level is defined with the help of a tree data structure (Aas et al., 2009)."
> > - Page 4, line ~208-209: the single quote right before the word "truncated" is in the wrong direction
> > - Page 4, equation (4): I'd suggest switching the equal sign with an approximation sign since once you truncate, it might not actually be equal
> > - Page 4, line ~215 right after equation (4): I don't understand precisely what the text here is saying; I'd suggest perhaps actually just formally writing it out.
> > - Page 5, line 219: "In this chapter" should say "In this section" (note that "chapter" also is used in the appendix one time where it should instead say "section"--look around line 758-759 of page 15)
> > - Page 5, line ~220: "in this first paragraph, we derive the final" -- I don't get why this part is phrased as "in this first paragraph" because I don't think what it's describing happens in this first paragraph
> > - Page 5, line ~224-225: "R^d" should say "\mathbb{R}^d" and the first quote for the word "normal" (i.e., the single quotation used right before the word) is in the wrong direction (also I would prefer that this particular sentence actually indicates that the samples drawn are i.i.d.)
> > - Page 5, line ~226-227: "if it violates any aspect of the distribution" -- I would reword this since what you end up checking against are very specific aspects and not just "any aspect" of the distribution. In other words, the marginal and dependence scores check for very specific notions of feature(s) taking on "extreme" values. There could be other aspects of features being "extreme" that are not captured by the specific marginal and dependence score functions used.
> > - Page 5, Equation (5): the notation is a bit clunky with the use of the definition operator ":=" twice; there are different ways to clean this up such as:
> > $$s_i = \frac{1}{d}\sum_{i=1}^d s_i\quad\text{where}\quad s_i := \frac{m_i + \sum_{j\in{CS_i}} d_{i \leftrightarrow j}}{1+\sum_{j\in{CS_i}} max_{i\leftrightarrow j}}$$

---

> > > ### Comment · Reviewer_5yhb · 2024-11-25
> > >
> > > - Page 5, line 253-254: fix the starting single quotes that are in the wrong direction (there are two occurrences, one in front of "minimal" and one in the front of "mass-volume"
> > > - Page 5, line 256: there is no need to put the transpose operator in stating $\boldsymbol{y}$
> > > - Page 5, line 257: previously in equation (2), $f_i$ was used for the density of the $i$-th variable whereas now it is $f_{X_i}$; I'd suggest being consistent in notation and separately, I'd also suggest being explicit when referring to a true distribution function or density vs referring to an estimator of a distribution function or density (for instance, now that the kernel density estimator is used, perhaps use the notation $\hat{f}_i$)
> > > - Page 5, lines ~262-263: I found this part confusingly written. At this point in the text, I'd suggest that you clearly motivate why precisely we must have $k = a^2$ (anticipating that the same grid will be used for pairs of inputs). Also in this line of text, I'd suggest also making it clear that you're inverting an empirical CDF (at least that's my interpretation of what is actually happening; maybe I'm missing something).
> > > - Page 6, line ~270-271: again, there is no need to put the transpose operator in stating $\boldsymbol{y}$
> > > - Page 6, around equation (7): again, I'd suggest being explicit with when estimators are used in place of true underlying copula functions that we do not actually know
> > > - Page 6, line ~282-283: is this standard usage of the word "pseudo-inverse"?
> > > - Page 6, line 286: I'd suggest rephrasing "We activate the estimated probability with a logarithm [...]" as "We take the logarithm of the estimated probability [...]"
> > > - Page 6, line 297: again, I find how the double usage of the definition operator to be used here to be clunky; perhaps use:
> > > $$d_{ij} := max_{ij} \cdot \big(-\log_b (p_{ij}(y))\big)\quad\text{where}\quad max_{ij} := sod_{ij} \cdot \eta^{t-1}$$
> > > - Page 6, last line: "we refer to the Appendix D.2" should instead say, for instance "we refer the reader to Appendix D.2" or "please see Appendix D.2"
> > > - Page 7, line 337: single quote before "normal" is in the wrong direction
> > > - Page 7, line 337: "$(0;1)$" should say "$(0,1)$"
> > > - Page 7, line ~352-353: "Else, if [...]" should say "Otherwise, if [...]" (Basically the use of the word "Else" here is awkward.)
> > > - Page 7, line ~360: single quote before "wine" is in the wrong direction
> > > - Page 9, lines ~449-453: fix beginning single quotes

---

### Official Review · Reviewer_JLN5 · 2024-11-04

**Soundness:** 3
**Presentation:** 3
**Contribution:** 2
**Rating:** 5
**Confidence:** 3

**Summary:**

The paper proposes a method for outlier detection that aims to achieve both interpretability and high accuracy. While powerful outlier detection methods using MLPs exist, they often lack explainability. This work, therefore, seeks to introduce a method that combines interpretability with high accuracy.

The method is based on Sklar's theorem, which decomposes the joint distribution of features into marginal distributions and a copula capturing dependencies between features. It employs kernel density estimation to estimate the univariate marginals and uses vine copulas to model the dependencies. Finally, the method assigns a score to each observation, classifying it as either normal or an outlier.

**Strengths:**

The strengths of this method lie in its ability to perform comparably to MLPs while providing interpretability by identifying which feature or feature dependency flagged an observation as an outlier. It is a simple method that nonetheless demonstrates good performance in experiments.

**Weaknesses:**

My primary concern is that the paper primarily combines existing methods and strategies rather than introducing new ones. It utilizes Sklar's Theorem, kernel density estimation, and vine copula methods—all of which have been previously developed. In this sense, the paper lacks sufficient novelty and originality. Its main contribution is limited to combining these established methods.

**Questions:**

The paper appears to consider a semi-supervised setting where only the labels of normal data are provided. Can this method be extended to cases where a few labeled samples from the outlier class are available (which could be viewed as a highly imbalanced classification problem)? How might this additional information enhance the method's effectiveness?

---

> ### Author Response · Authors · 2024-11-17
>
> We thank the reviewer for taking the time to read the paper, and for identifying potential shortcomings. Below, we respond to the question and the point of missing novelty.
>
> ## Weaknesses
>
> 1. *My primary concern is that the paper primarily combines existing methods and strategies rather than introducing new ones. It utilizes Sklar's Theorem, kernel density estimation, and vine copula methods—all of which have been previously developed. In this sense, the paper lacks sufficient novelty and originality. Its main contribution is limited to combining these established methods.*
>
> While it is true that VC-BOD combines several existing statistical methodologies, we would like to emphasize the key novelties of our contribution:
>
> - Novel Outlier Detection Method: VC-BOD introduces a new approach to outlier detection.
> - Explainability Framework: We propose an innovative explainability framework grounded in theory, focusing on feature attribution and distinguishing whether identified outliers result from violations in marginal distributions or dependencies between features.
>
> To the best of our knowledge, our work is the first to explicitly separate marginal feature distributions from dependencies between features in the context of outlier detection. This novel aspect is crucial to our method’s ability to achieve performance comparable to state-of-the-art deep learning models while surpassing other purely statistical approaches.
>
> ## Questions
>
>
> 1. *The paper appears to consider a semi-supervised setting where only the labels of normal data are provided. Can this method be extended to cases where a few labeled samples from the outlier class are available (which could be viewed as a highly imbalanced classification problem)? How might this additional information enhance the method's effectiveness?*
>
>
> We greatly appreciate this question. VC-BOD can indeed be adapted from a zero-shot to a few-shot setting. Conceptually, this adaptation could involve penalizing the density in regions where the few known outliers are observed. If these outliers stem from a distinct distribution that is significantly different from the normal data, it is reasonable to expect that VC-BOD could perform effectively in a few-shot scenario.
>
> However, implementing this extension would require significant modifications to the underlying statistical framework, including adjustments to kernel density estimation, methodology, and the experimental setup. Given the scope and complexity of this exploration, we believe this promising direction deserves a dedicated project to ensure a comprehensive and rigorous investigation.
>
>
>
> We hope this response adequately addresses the reviewer’s concerns. We invite the reviewer to engage in further discussion regarding our paper.

---

### Official Review · Reviewer_b1G3 · 2024-11-04

**Soundness:** 2
**Presentation:** 2
**Contribution:** 3
**Rating:** 6
**Confidence:** 3

**Summary:**

This paper focuses on outlier detection. The authors apply Sklar’s theorem to approximate the distribution of normal observations via marginal distributions and a copula. Based on the approximated distribution, an outlier detection method is proposed by computing the scores of marginals and pair-copulas. The proposed method is able to explain the decision of detection, the decisive features, and the type of decisive features. Experimental results show that their method surpasses other classical methods.

**Strengths:**

- The most significant strength of this paper is the interpretability of their method. The interpretability of this method is three-fold, in terms of explaining the reason of decision, the features that contribute to decision, and the type of decisive features, which is still a gap in deep learning-based approaches.
- Compared to other deep learning-based approaches, another benefit of the method is its efficiency. It can be applied with CPUs in seconds on most datasets, which is much faster than training a Transformer. While the scalability of this method is not clear yet, it has a low computation cost in small datasets.

**Weaknesses:**

- Some notations in this paper is not clear with some potential typos, and some technical details are not well explained. I recommend the author to improve the presentation and include necessary explanation to make it more readable. Please check **Questions** for more details.
- Most datasets used in experiments are in a small dimension, which limits the impact of the proposed method. In Tab. 6, it takes several hours to run on larger datasets, such as Mullcross and Forest. I think it would be better if the author can discuss the scalability of the method with larger datasets.
- This paper mainly applies the Sklar’s theorem but lacks some necessary discussions, such as an ablation study of hyperparameters and the assumptions on normal distributions. Given that there is no theoretical analysis of the influence of those hyperparameters and distribution assumptions, it would be better for the authors to discuss it thoroughly to enhance the impact and practicability of their method.

**Questions:**

- In line 165, $u_i$ is defined as $(F_1(x_1^i),...,F_d(x^i_d))$, while its definition in Eq. 3 is confusing to me.
- In Eq. 4, what does $c_{i,j}$ represent? I don’t understand what is $F_i(x_i|d_{i,j})$ as well. What is $F_i$ conditioned on? How does $F_i$ concatenate the transformation of the feature?
- What is the effect of hyperparameters such as $k$ and $\eta$ on the performance? The author uses a default choice of those hyperparameters but there is a lack of guidance on how to selec them.
Minor:
- In lin 143, $[0;1]$ -> $[0,1]$
- In line 317, $f_2(d)$ -> $f_2(n)$

---

> ### Author Response · Authors · 2024-11-18
> **Response to Weaknesses**
>
> Dear reviewer, first of all, thank you for your positive feedback, reading the manuscript carefully,
> raising important questions and pointing out formatting issues. A reply to your questions and suggestions can be found below.
>
> ## Weaknesses
>
> 1. *Some notations in this paper is not clear with some potential typos, and some technical details are not well explained. I recommend the author to improve the presentation and include necessary explanation to make it more readable. Please check Questions for more details.*
>
> We thank the reviewer for pointing out the typos. We corrected the typos, and hope to have improved the mentioned explanations.
>
> 2. *Most datasets used in experiments are in a small dimension, which limits the impact of the proposed method. In Tab. 6, it takes several hours to run on larger datasets, such as Mullcross and Forest. I think it would be better if the author can discuss the scalability of the method with larger datasets.*
>
> While this problem does exist, we want to mentioning that it is not a theoretical problem but rather a problem caused by the current implementation of the copula methods, which only work on CPUs. Large datasets could likely benefit from an implementation utilizing GPUs.
>
> 3.  *This paper mainly applies the Sklar’s theorem but lacks some necessary discussions, such as an ablation study of hyperparameters and the assumptions on normal distributions. Given that there is no theoretical analysis of the influence of those hyperparameters and distribution assumptions, it would be better for the authors to discuss it thoroughly to enhance the impact and practicability of their method.*
>
> We appreciate the reviewer pointing out that the implicit assumptions made by VC-BOD were not discussed in detail in the paper. We have, also in response to a different review, added a thorough statistical analysis of VC-BOD where we explicitly formulate the assumptions of VC-BOD:
>
> - (A1) The training sample consists of i.i.d. realizations from an unknown distribution.
> - (A2) The unknown distribution allows for a vine copula specification, which can be determined using Dissmann's algorithm with Gaussian kernels for the marginals and Gaussian kernel copulas for the pair-copulas, where
>     - (A2-1) The marginal distributions are correctly specified.
>     - (A2-2) The pair-copulas in the trees $T_i$ with $i=2,\dots,T$ are independent of the conditioning variables.
>     - (A2-3) The pair-copulas in the trees $T_i$ with $i=1,\dots,T$ are correctly specified.
>
>
>
> For more details, we kindly refer the reviewer to the newly added Section A.1 of the Appendix.
>
>
> Concerning the hyper-parameters, we claim that the central hyper-parameter of VC-BOD is the truncation level $T$. We detailed the theoretical influence of this hyper-parameter in the main text in Section 4 and, in greater detail, in the third paragraph of the newly added Section A.1 of the Appendix. The empirical influence is studied through the comparison of the version VC-BOD T0 - T3 in the empirical study of the paper.
>
> The hyper-parameter $k$ governs the quality of the approximation of the integrals in equations (6) and (7) in the main text. The standard convergence rate for Monte-Carlo type approximations is $O(k^{-0.5})$. In our setting, we use stratified sampling, which can be assumed to speed up this rate, given regularity assumptions.
>
> The hyper-parameter $\eta$ is a regularization parameter intended to compensate for possible violations of the simplifying assumption (A2-2) for trees $T>1$. For ${\scriptsize \eta \to 0}$, VC-BOD T3 converges to VC-BOD T1 as the influence of feature interactions captured in higher trees diminishes. The performance of VC-BOD for other choices of $\eta$ highly depends on how well the assumptions, especially (A2-2), are fulfilled.

---

> ### Author Response · Authors · 2024-11-18
> **Response to Questions**
>
> ## Questions
>
> 1. *Question 1.*
>
> We apologize for the confusion. The correct definition is as follows.
>
> $ u^{(k)} = (u^{(k)}_1, \dots, u^{(k)}_d) = (F_1(x_1^{(k)}), \dots, F_d(x_d^{(k)}))$
>
> 2. *Question 2.*
>
> We appreciate the reviewer pointing out the potential confusion. We clarified the notation below:
> $c_{i,j}$ denotes the density of the pair copula which connects features $i$ and $j$. In $F_i(\cdot| \boldsymbol{d}_{i,j})$, the following transformations are concatenated: First, the probability integral transform converts the i-th feature to the range $[0,1]$ by applying the i-th marginal distribution, see line 165. Second, the i-th feature is transformed to match the conditioning of the pair copula, i.e., if the pair copula connects feature 1 and 3, given 2, then $u_1$ and $u_3$ need to be conditioned on $u_2$.
>
> The sentence from the paper 'where $F(...)$ concatenates the transformation of the feature to the domain of the copula with the internal transformations of the vine-copula' could be replaced by: 'where $F(...)$ concatenates two transformations: First, the feature is transformed to $[0,1]$ by applying the marginal distribution function to it, and second, the point is transformed according to the previous trees in the vine-copula to match the conditioning $d_{i,j}$ of the corresponding pair-copula.'
>
> We kindly ask the reviewer to assess whether this substitution would improve readability.
>
> 3. *Question 3.*
>
> We kindly refer the reviewer to our answer to Weakness 3.
>
>
> 3. *Question 4&5.*
>
> We want to thank the reviewer for pointing out the minor issues with the formatting, which we addressed in the revised version.

---

> > ### Comment · Reviewer_b1G3 · 2024-11-27
> > **Thanks for the rebuttal**
> >
> > Thanks for the authors' response. While most of my questions are addressed, I am still concerned about the scalability and practicability of this method. It is not clear whether this method can be used efficiently in a larger dataset in terms of the number of both the dataset size and feature dimension. Also, the effect of different hyperparameters has not been studied empirically. Thus, I would like to keep my score and encourage the authors to conduct more experiments on larger datasets to evaluate the impact of hyperparameters.

---

### Official Review · Reviewer_BLV7 · 2024-11-04

**Soundness:** 3
**Presentation:** 3
**Contribution:** 3
**Rating:** 3
**Confidence:** 4

**Summary:**

The authors introduce a copula based outlier detection method, that focus on fully utilizing the Sklar's Theorem (the density of a random vector can be written as a factor of its univariant marginals and the gradient of an appropiate copula). The authors utilizes a Vine Copula, and fit all the densities using gaussian kernels with the ROT [Sheather & Jones] bandwidth. They provide a unique way of characterizing the outlier score of an outliying sample by penalizing the non-complience of any of the factor's inlier condition (in eq. 5). Furthermore, the authors provide a way of assigning explainability of an outlier sample using the Sklar's theorem formulation.
They tested their One-class classification performance in section 6 against several classical and DL methods using a collection of 31 datasets across 40 repetitions. The Vine Copula-Based Outlier Detection method (VC-BOD) that they presented is claimed to achieve a significant performance over the collection of classical methods, while providing a competitive performance againts the more complex Deep Learning ones.

**Strengths:**

1. The model's novelty is good, and the idea of fully utilizing the Sklar's theorem is sounded
2. The motivation of the paper is solid and well presented by the authors
3. The method seem to provide a proper outlierness explanation following their experiments in section B.1. Explainability of outlier detection models is crucial for industrial-level applications.
4. The experiment section utilizes a fair amount of datasets and a large collection of competitors
5. The authors provide the code for reproducibility
6. The authors also provide a theoretical complexity analysis.

**Weaknesses:**

1. The model lacks a proper theoretical background to explain why it should work. While it is important to remark that the explainaitions are grounded in theory, the authors do not provide any guarantee of classifying an outlier as an outlier. The authors have access to an aproximation of the joint density function (as per the Sklar's theorem), which could solve this weakness. However, they do not seem to use it.

2. The experimental evaluation is poor (which contributes to my score in Soundness). While it is true that the authors make use of a large list of datasets with a sound experimental setting, they do not use any statistical test to assest the significance of their results. Without this, it is impossible to assest wether their claims of SotA performance are true. The importance of these tests are highlighted throught recent OD literature [Campos et.al.] [Han et.al.] [Liu et.al], and even outside of Outlier Detection, even being a course of reject for similarly-sized conferences to ICLR (see section 7 of NeurIPS submission checklist).

3.The paper has a theoretical complexity study, and yet, there is no experimental evaluation of it.

4.There is no experimental analysis of the explaination score with real world datasets, even though it is one of the major selling points of the method.

**Questions:**

I ask the authors to please answer my following questions:

1. Why did they authors implement a different scoring function that the joint density?

2. There exists a vact collection of xAI Outlier Detection methods in the literature [Sejr and Schneider-Kamp]. Is there any reason as to why the authors did not compare themselves to other xAI competitors?

3. Lines 298-300 contains the following claim "If the vine copula fit is close to the ground truth, then adding more trees increases accuracy by reducing the number of false negatives,(...)". Why this would happend utilizing the scoring presented in eq. (5) is not imediate to me. Additionally, it seems to be an extreamly important theoretical result for your model that should be highlighted more.

4. Please consider introducing a multiple comparison test into the experimental evaluation. As it is currently, it is impossible to assest if the method is not worst that the competitors presented.

I will consider increasing my score after succesfully addressing all of my questions/concerns listed above.

### References
 [Sheather & Jones] Sheather, S. J., & Jones, M. C. (1991). A Reliable Data-Based Bandwidth Selection Method for Kernel Density Estimation. Journal of the Royal Statistical Society. Series B (Methodological), 53(3), 683–690. http://www.jstor.org/stable/2345597

 [Campos et.al.] Guilherme O. Campos et.al. (2016). On the evaluation of unsupervised outlier detection: measures, datasets, and an empirical study. Data Mining and Knowledge Discovery 30, 891-927

[Han et.al.] Han, S., Hu, X., Huang, H., Jiang, M., & Zhao, Y. (2022). ADBench: Anomaly Detection Benchmark. In Advances in Neural Information Processing Systems (pp. 32142–32159).

[Liu et.al] Yezheng Liu et.al. (2020) Generative Adversarial Active Learning for Unsupervised Outlier Detection. IEEE TRANSACTIONS ON KNOWLEDGE AND DATA ENGINEERING, VOL. 32, NO. 8, AUGUST 2020.

 [Sejr and Schneider-Kamp] Jonas Herskind Sejr, Anna Schneider-Kamp, Explainable outlier detection: What, for Whom and Why?, Machine Learning with Applications, Volume 6, 2021, 100172, ISSN 2666-8270, https://doi.org/10.1016/j.mlwa.2021.100172.

---

> ### Author Response · Authors · 2024-11-18
> **Response to Weaknesses**
>
> Dear reviewer, first of all, thank you for reading our manuscript carefully and raising important
> questions. We hope we could address your concerns in the revised version of the paper. In the following we first address the mentioned weaknesses.
>
> ## Weaknesses
>
> 1. *The model lacks a proper theoretical background to explain why it should work...*
>
>
> We have added a detailed theoretical derivation of VC-BOD in the newly included Section A.1 of the appendix. We kindly invite the reviewer to refer to this section in the updated version of the paper, as the detailed explanation is too lengthy to include in this response.
>
> To summarize, the newly added section frames VC-BOD within the context of multiple testing. When outlier detection is interpreted as out-of-distribution detection, and under appropriate assumptions, we argue that VC-BOD’s outlier detection task can be framed as a multiple hypothesis testing problem. Specifically, VC-BOD evaluates whether a new observation aligns with every aspect of a learned distribution based on the sample. These aspects are represented by the marginals and the pair-copulas of the vine copula. If sufficient evidence emerges that one or more aspects are violated, the observation will be classified as an outlier.
>
> Regarding the second point raised about the use of the joint density function, we hope to have addressed this thoroughly in our response to the reviewer’s first question to avoid redundancy.
>
> 2. *The experimental evaluation is poor (which contributes to my score in Soundness)....*
>
> We appreciate the reviewer’s emphasis on the importance of statistical significance testing in experimental evaluations. Following the guidance from the cited literature [Campos et al.; Han et al.; Liu et al.], we conducted a Friedman test coupled with a Nemenyi post-hoc test to evaluate the significance of our ranking claims. The detailed results can be found in in the newly added Appendix Section D.1.
>
> In summary, our statistical analysis highlights several key findings.
> VC-BOD outperforms several baseline models, including COPOD, PIDForest, RRCF, and NeuTraL. Additionally, our analysis indicates no statistically significant difference between VC-BOD and KNN, nor between VC-BOD and leading deep learning approaches such as NPT-AD. These findings further highlight VC-BODs postition at the forefront in terms of performance.
>
>
>
> 3. *The paper has a theoretical complexity study, and yet, there is no experimental evaluation of it.*
>
> For our complexity analysis, there are two components: 1) estimate the vina copula and 2) calculate the outlier score of the sample values.
>
> The complexity analysis of the first part has already been addressed in prior research, specifically in the VCAE paper [Tagasovska et al., 2019](https://arxiv.org/abs/1906.05423) in Section 2.3 on page 4. This previous work also implemented truncated vine copulas using a combination of kernel density estimators for the marginal distributions and Gaussian kernel copulas for the pair-copulas.
>
> In our case, an additional second step involves calculating the sample outlier score for each component. This task reduces, for each individual outlier score, to sorting a vector of length $n$ into a vector of length $k$. The complexity of this operation is $f_3(n, k) = O((n + k) \cdot \log(k))$ (note: there was a typo in this formula in the previous draft). This computation is added to the first part of the complexity, which pertains to the estimation of the vine copula.
>
> Due to limited computational resources, we are unable to conduct an extensive empirical evaluation of the complexity independently.
>
> 4. *There is no experimental analysis of the explaination score with real world datasets, even though it is one of the major selling points of the method.*
>
> We kindly ask the reviewer for some additional time to address this point.
>
>
> [Campos et.al.] Guilherme O. Campos et.al. (2016). On the evaluation of unsupervised outlier detection: measures, datasets, and an empirical study. Data Mining and Knowledge Discovery 30, 891-927
>
> [Han et.al.] Han, S., Hu, X., Huang, H., Jiang, M., & Zhao, Y. (2022). ADBench: Anomaly Detection Benchmark. In Advances in Neural Information Processing Systems (pp. 32142–32159).
>
> [Liu et.al.] Yezheng Liu et.al. (2020) Generative Adversarial Active Learning for Unsupervised Outlier Detection. IEEE TRANSACTIONS ON KNOWLEDGE AND DATA ENGINEERING, VOL. 32, NO. 8, AUGUST 2020.
>
> [Tagasovska et al., 2019](https://arxiv.org/abs/1906.05423) Tagasovska, Natasa, Ackerer, Damien, and Vatter, Thibault. “Copulas as High-Dimensional Generative Models: Vine Copula Autoencoders.” arXiv preprint, 2019.

---

> ### Author Response · Authors · 2024-11-18
> **Response to Questions**
>
> ## Questions
>
> 1. *Why did they authors implement a different scoring function that the joint density?*
>
>
> Utilizing the entire density would require more restrictive assumptions—specifically, that the entire vine copula is correctly specified. This a very serve and in practice often times inappropriate assumption. In our setting, we truncate the vine copula after tree number 0, 1 or 3 and implicitly assume correctness only up to that point. For a more detailed explanation, we kindly refer to the first paragraph of the new added Section A.1 in the Appendix, where the implicit assumptions of VC-BOD are discussed.
>
> Moreover, it is established that outlier detection suffers from the curse of dimensionality [Zimek et al., 2012](https://doi.org/10.1002/sam.11161), posing both theoretical and practical challenges.
>
> While it would be possible to have explainability with the entire density to some extent, we constructed our scoring method with our explainability framework in mind, which necessitated constructing a new functional form for the outlier score.
>
> 2. *There exists a vact collection of xAI Outlier Detection methods in the literature [Sejr and Schneider-Kamp]. Is there any reason as to why the authors did not compare themselves to other xAI competitors?*
>
>
> Using the nomenclature of [Sejr and Schneider-Kamp](https://doi.org/10.1016/j.mlwa.2021.100172), our method is "directly interpretable" and falls under the category of "subspace methods," under the broader category of “white box explanations”, which is presented in their work to be the most straight forward form of interpretability. In contrast, many existing advanced methods are classified as black box methods, which do not offer direct interpretability.
>
> For explainability to be meaningful, we claim that a model must also perform at a sufficiently high level in terms of accuracy.
> Therefore, having established our model's explainability framework, we then focused on comparing our method with the most advanced and competitive approaches in current research. Specifically those published at leading conferences like ICLR and ICML in recent years, such as [Thimonier et al., 2024](https://arxiv.org/abs/2406.05423).
>
>
>
> 3. *Lines 298-300 contains the following claim "If the vine copula fit is close to the ground truth, then adding more trees increases accuracy by reducing the number of false negatives,(...)"....*
>
>
> We thank the reviewer for pointing out that this part was not detailed sufficiently. A detailed explanation of this tradeoff is provided in the third paragraph of the newly added  Section A.1 in the Appendix, it can be summarized as follows:
>
> With an increasing truncation level, we check more and more sufficient conditions for a new observation to be considered an outlier, which increases the chances of finding suitable conditions for demasking true outliers.
>
> An example for this is the empirical example in Section 6.1 of the main paper.
>
>
>
> 4. *Please consider introducing a multiple comparison test into the experimental evaluation. As it is currently, it is impossible to assest if the method is not worst that the competitors presented.*
>
> This point is addressed within our rebuttal to weakness number 2.
>
>
> We hope this response adequately addresses the reviewer’s concerns, especially regarding weaknesses 1 and 2. We invite the reviewer to engage in further discussion of our paper.
>
>
> [Zimek et al., 2012](https://onlinelibrary.wiley.com/doi/abs/10.1002/sam.11161) Zimek, Arthur, Schubert, Erich, and Kriegel, Hans-Peter. "A survey on unsupervised outlier detection in high-dimensional numerical data." *Statistical Analysis and Data Mining: The ASA Data Science Journal*, 5(5), 363–387, 2012.
>
>
> [Sejr and Schneider-Kamp](https://doi.org/10.1016/j.mlwa.2021.100172) Jonas Herskind Sejr, Anna Schneider-Kamp, Explainable outlier detection: What, for Whom and Why?, Machine Learning with Applications, Volume 6, 2021, 100172, ISSN 2666-8270,
>
>
> [Thimonier et al., 2024](https://arxiv.org/abs/2305.15121) Thimonier, Hugo, Popineau, Fabrice, Rimmel, Arpad, and Doan, Bich-Liên. “Beyond Individual Input for Deep Anomaly Detection on Tabular Data.” arXiv preprint, 2024.

---

> ### Author Response · Authors · 2024-11-20
> **Reponse to Weakness 4.**
>
> 4. *There is no experimental analysis of the explaination score with real world datasets, even though it is one of the major selling points of the method.*
>
> We have included an experimental analysis of our proposed explainability framework in the newly added Section B.3 of the Appendix. This experiment highlights the practical applicability of VC-BOD, particularly through its diagnostic plots, which provide a clear and intuitive way to analyze and understand why specific observations are labeled as outliers.
>
> Having thoroughly addressed all the weaknesses and questions raised by the reviewer, we look forward to receiving the reviewer's feedback and the opportunity to further discuss the paper.

---

> ### Comment · Reviewer_BLV7 · 2024-11-26
> **Reviewer's Response**
>
> Dear authors. I appreciate the effort made during the rebuttal to answer my question and tackle all the weaknesses I've raised. My response will be organized as follows. I will first respond to all of your points raised. I will then summarise my conclusions. To finalize, I will give my final decision regarding the score to the AC.
>
> ## Answers
>
> ### Weaknesses
>
> W1. This comparison is unnecessary. The point I raised was that the method lacked any theoretical motivation as to why it should work. The authors responded to this by framing the method as a multiple-test comparison. However, this does not refute my point, particularly because, I quote, "(...) assuming the individual p-values to be independent is not reasonable. As a result, the distribution under [the null] is not known". I.e., the theoretical derivations one can obtain by using this proposed framework (consistency of the test, power, boundings for the errors) that could motivate its performance and their explanations, *are not applicable in this context*. My point about how there is no theoretical motivation (nor for its explanations, and its performance) still holds.
>
> W2. I appreciate the correction. Right now, the test confirms that VC-BOD cannot outperform baselines like IForest, KNN, and other deep-learning methods. Granted, it is worth mentioning that VC-BOD can attribute a higher level of interpretability to its results than the other Deep Methods. However, the test is not correctly performed, and it could use a rework for later revisions of the manuscript (see Questions).
>
> W4. One dataset and no evaluation with other xAI competitors does not answer the question of "Why should I use this method in the context of xAI".
>
> ### Questions
>
> Q1. This reasoning seems arbitrary. Fisher's Method of Combining Tests assumes independence of all tests [Little & Folks]. The authors also mention that this is something completely unreasonable in this setting (L857-L858).
> Therefore, the authors failed to show that the statistic used for the test has any guarantee of correctly assessing the null hypothesis on a given critical level. Given this, there seems to be no reason as to why one would prefer to combine each test than to use the approximation of the joint density.
>
> Q2. I do agree with the necessity of being competitive in terms of accuracy. However, if the proposed use case is xAI in outlier detection, the method not only has to be compared against other ODM, but also the other ODM for xAI. Additionally, one also needs a proper review of the RW in the field to know the specific advantages of VC-BOD over other xAI methods.  Without this, VC-BOD cannot be positioned as a valid competitor in the field. Right now, due to these points and the points raised in (Q.1), there are serious doubts about the use of VC-BOD as an xAI method.
>
> Q4. In an unsupervised setting, different versions of the models should be compared individually. When using an MCT, adding more pair-wise tests reduces the power of the multiple-comparison test. All of the VC-BOD versions should be, either,
>    A] Compared individually with the rest (resulting in 3 versions of the test)
>    B] Compare the best and worst performing versions to know the best and worst case scenarios.
>
> ## Summary
>
> As a summary of my initial comment, I had 2 main questions about the manuscript.
> 1. Why should I use this method as an ODM?
> 2. Why should I care about the outlier explanations that this method gives?
>
> After the response, these questions have not been answered.
> 1. There is no theoretical motivation for this method that tells me why It should perform better than other ODMs or xAI-ODMs. Additionally, the experimental results return that this method failed to improve over established and well-known baselines like IForest or KNN. (See Answer W2)
> 2. There is no experimental, nor theoretical motivation as to why someone could prefer this method over other xAI-ODMs. In fact, there is (to my knowledge) no in-text review of the current state of xAI for OD. Without this, it is impossible to position this method in the field. (See Answers Q1, W1, Q2, W4).
>
> ## Final Score
>
> I will increase my score in soundness since the authors successfully addressed the issue of the lack of a statistical comparison of the results. (1-->3). I will maintain my suggestion to reject as I believe that the authors failed to motivate the use of this method over several other alternatives (3).

---

### Meta-Review · Area_Chair_2Yr4 · 2024-12-19

**Metareview:**

Based on the reviews, I conclude that the paper cannot be accepted for publication in its current form. Of the five reviews, four recommend rejection. The reviewers raised significant concerns regarding key aspects of the work, including the lack of comparative analysis, insufficient theoretical depth, and weak empirical evaluation.

**Additional Comments On Reviewer Discussion:**

Key points raised:

- **Theoretical and Empirical Evaluation**: Reviewer BLV7 noted a lack of theoretical guarantees and weak empirical evaluation. The authors addressed some issues but failed to improve over baselines or provide sufficient theoretical motivation, leading to a rejection recommendation.

- **Novelty**: Reviewer JLN5 questioned the paper’s novelty, while Reviewer BLV7 listed it in strengths. The authors responded, but JLN5 did not change their rating.

- **Scalability and Technical Details**: Reviewer b1G3 raised issues with scalability and technical clarity. The authors added statistical analysis, but concerns about scalability remained.

- **Empirical Setup**: Reviewer 5yhb flagged weaknesses in the empirical evaluation. The authors clarified their setup, but issues persisted, leaving the rating unchanged.

- **Complexity and Assumptions**: Reviewer adkm raised concerns about computational complexity and assumptions, which the authors partially addressed.

In my final decision, concerns about theoretical depth, empirical evaluation, and scalability outweighed the revisions, leading to the above recommendation.

---

### Decision · Program_Chairs · 2025-01-22

Reject